# Developing Pharmacological Therapies for Atrial Fibrillation Targeting Mitochondrial Dysfunction and Oxidative Stress: A Scoping Review

**DOI:** 10.3390/ijms25010535

**Published:** 2023-12-30

**Authors:** Antônio da Silva Menezes Júnior, Ana Luísa Guedes de França-e-Silva, Joyce Monteiro de Oliveira, Daniela Melo da Silva

**Affiliations:** 1Internal Medicine Department, Medicine School, Federal University of Goiás, Goiânia 74605-020, GO, Brazil; ana.guedes@discente.ufg.br (A.L.G.d.F.-e.-S.); danielamelosilva@ufg.br (D.M.d.S.); 2Medicine Department, Medical and Life School, Pontifical Catholic University of Goiás, Avenida Universitária, 1440, Sector Universitario, Goiânia 74605-010, GO, Brazil; antonio.med@puc.br

**Keywords:** oxidative stress, atrial fibrillation, mitochondrial dysfunction, inflammation, antioxidant drug discovery

## Abstract

Atrial fibrillation (AF) is a cardiac arrhythmia caused by electrophysiological anomalies in the atrial tissue, tissue degradation, structural abnormalities, and comorbidities. A direct relationship exists between AF and altered mitochondrial activity resulting from membrane potential loss, contractile dysfunction, or decreased ATP levels. This review aimed to elucidate the role of mitochondrial oxidative mechanisms in AF pathophysiology, the impact of mitochondrial oxidative stress on AF initiation and perpetuation, and current therapies. This review followed the Preferred Reporting Items for Systematic Reviews and the Meta-Analysis Extension for Scoping Reviews. PubMed, Excerpta Medica Database, and Scopus were explored until June 2023 using “MESH terms”. Bibliographic references to relevant papers were also included. Oxidative stress is an imbalance that causes cellular damage from excessive oxidation, resulting in conditions such as AF. An imbalance in reactive oxygen species production and elimination can cause mitochondrial damage, cellular apoptosis, and cardiovascular diseases. Oxidative stress and inflammation are intrinsically linked, and inflammatory pathways are highly correlated with the occurrence of AF. AF is an intricate cardiac condition that requires innovative therapeutic approaches. The involvement of mitochondrial oxidative stress in the pathophysiology of AF introduces novel strategies for clinical treatment.

## 1. Introduction

Atrial fibrillation (AF) is the most frequent and clinically significant arrhythmia and a substantial risk factor for stroke and heart failure (HF) [1]. It describes an abnormal pattern of electrical activity in the atrium, which results in inefficient contraction of the atrium [2]. More than 33 million individuals are affected by AF, and the likelihood of having AF rises with age [3,4]. Electrophysiological and atrial remodeling alterations are believed to be the primary causes of the development of paroxysmal to persistent AF [5]. AF evolution is driven by atrial fibrosis, which is a hallmark of atrial structural remodeling [6]. Although clinical studies of AF have significantly advanced, effective therapeutic techniques are still needed to prevent it [7].

As the heart is an energy-starved organ, an adequate source of cardiac energy is necessary for appropriate functioning [8]. Electrical and mechanical remodeling of the heart caused by AF may significantly impair energy metabolism, mainly through glucose and lipid metabolism [9,10]. Several studies have shown a reduction in ATP generation in the atrial tissues of patients with AF, and mice treated with angiotensin (Ang) II have been demonstrated to indicate unreliable energy metabolism in the atrium [11,12]. Mitochondria are important organelles responsible for creating energy and are considered crucial participants in the energy metabolism and redox status of the myocardium [13,14]. The development of AF may be accompanied by mitochondrial dysfunction, which can result in inadequate ATP production and the buildup of reactive oxygen species (ROS), further disrupting [15].

Recent studies have shown that inflammation and oxidative stress play pivotal roles in the structural and electrical alterations of the atrial tissue, potentially contributing to the pathogenesis of AF [16,17]. Infiltration of inflammatory cells and calcium accumulation during episodes of heightened atrial rhythm in AF may cause oxidative damage within the atrial myocardium [18]. This, in turn, may expedite the development of atrial fibrosis, thereby facilitating its progression. The predominant generators of ROS within atrial tissue are the mitochondrial electron transport chain, xanthine oxidase, uncoupled nitric oxide synthase (NOS), and nicotinamide adenine dinucleotide phosphate (NADPH) oxidases [19]. Xanthine oxidase is posited to play an instrumental role in redox signaling across various cardiovascular disorders, particularly emphasizing its involvement in AF (Figure 1) [20,21,22,23,24,25].

Noninvasive therapies for AF are still unclear. However, several medications are effective and safe for treating either the preventive or definitive forms of the disease [10]. However, some drugs have limited efficacy and pose considerable risks to patients. Moreover, these therapies are more related to electrical than structural disturbances [5]. In addition, there are combination pharmacotherapies that can treat AF, such as drugs prescribed for diabetes mellitus (DM) and other disorders, including dipeptidyl peptidase-4 inhibitors, selective sodium-glucose cotransporter inhibitors 2, ubiquinone, metformin, thiazolidinediones, fibrates, trimetazidine, and ranolazine [4].

Antiarrhythmic therapies include pharmacological agents for rhythm and rate control and catheter-based interventions, which are mainly effective on paroxysmal AF and less effective on permanent AF. Both effectively prevent secondary episodes of AF, but they are still far from treating all forms [6,7]. Therefore, a better understanding of AF mechanisms is urgently required to develop more effective therapeutic approaches. However, we aimed to correlate the role of mitochondrial oxidative stress and point out the role of ROS in the genesis of AF to elucidate recent promising pharmacological therapies.

## 2. Methods

### 2.1. Type of Study

This study is a scoping literature review based on the stages proposed by the Preferred Reporting ltems for Systematic Reviews and Meta-Analysis for Scoping Reviews. The final protocol was prospectively registered in the Open Science Framework on 28 November 2023. Electronic databases such as PubMed, Excerpta Medica Database, and Scopus were searched up to October 2023 utilizing the combination of MESH terms “Oxidative Stress”, “Mitochondrial”, and “Atrial Fibrillation”. Additionally, bibliographic references were manually checked for relevant articles included in this review.

### 2.2. Review Question

The review question was formulated using the PCC strategy. Problem: AF; Concept: Oxidative stress and mitochondrial dysfunction; Context: Emerging pharmacological therapeutic possibilities. Thus, the review question was: “How do oxidative stress and mitochondrial dysfunction modulate atrial fibrillation from the perspective of therapeutic possibilities?”

### 2.3. Protocol and Registration

This review was registered in the Open Science Framework (DOI registry 10.17605/OSF.IO/VSZ72). 

### 2.4. Eligibility Criteria

Inclusion criteria: This review includes primary and secondary studies that evaluated genetic changes and/or mitochondrial dysfunction in hypertrophic cardiomyopathy (among other variables, such as oxidative stress). The review encompassed a variety of study designs, including case-control studies, reviews, clinical studies, experimental studies, randomized controlled trials, cross-sectional studies, and prospective studies.

Exclusion criteria: Excluded from consideration in the review were secondary sources such as editorials, books, expert opinion articles, dissertations, theses, and conference abstracts, except for literature reviews, which were included.

### 2.5. Sources of Information and Search Strategy

Studies were searched in the following databases: Excerpta Medica Database, SciVerse Scopus, and PubMed. The search strategy was formulated from a combination of controlled descriptors and/or keywords related to the topic without applying restrictions related to the language of the publication period. In addition, a manual search was conducted on the reference lists from the initially selected studies to identify other eligible studies.

### 2.6. Process of Study Selection

The identified studies were imported into Rayyan software, and duplicates were removed. Studies without duplicates were evaluated and selected based on eligibility criteria by three independent and blinded reviewers by reading the titles and abstracts of the studies (A.S.MJ, H.V., and A.L.), followed by reading the full text of the selected studies in phase 1. A fourth reviewer solved disagreements in this study selection process (D.A.M.).

### 2.7. Process of Data Extraction from Selected Studies

The data from the selected studies was rigorously analyzed and collected by three independent and blinded reviewers by filling out a characterization table in Microsoft Word Software 365 version, which contains the characteristics of this study: identification (citation), study design, and country where the study was developed; aspects of atrial fibrillation pathophysiology: concept, mitochondrial dysfunction, oxidative stress, and metabolic disorders; and primary outcome: emerging pharmacological therapeutic possibilities.

### 2.8. Risk of Bias Assessment or Quality Assessment

Because this scoping review was conducted to identify knowledge gaps, there was no risk of bias or quality assessment, according to the manual published by the Joanna Briggs Institute.

### 2.9. Data Synthesis

A qualitative synthesis of the selected studies’ data are provided, describing the genetic mutation and mitochondrial dysfunction proposed in each study according to the etiology of hypertrophic cardiomyopathy, whether it originates from a genetic mutation (sarcomeric or mitochondrial) or dysfunctions in cellular metabolism, among others. A descriptive table summarizes all this information.

## 3. Results

Among the 2175 articles in the initial search, 37 were selected for the final analysis (Figure 2). Primary data from each study are presented in Table 1. The main categories of analysis were mitochondrial dysfunction and oxidative stress modulators, focusing on new therapeutic possibilities.

### 3.1. Oxidative Stress

#### 3.1.1. Mitochondrial Dysfunction

Oxidative phosphorylation occurs when electrons from the transport chain react with oxygen molecules to generate ROS. Mitochondria are a significant source of ROS in cardiomyocytes, with the average production balanced by cellular antioxidant mechanisms [63]. Increased ROS levels can activate calcium-dependent transcription factors. However, sustained and uninhibited increases can lead to mitochondrial damage, mitochondrial DNA damage, and damage to proteins and lipids. High ROS levels can impair creatine kinase and disrupt the balance between ATP and ADP. Oxidative damage to mitochondrial DNA coincides with a decrease in the energy-producing capacity of the heart mitochondria, leading to an increase in ROS release. This can lead to AF pathology, associated with electrophysiological, contractile, and structural remodeling [63]. Oxidative stress (OS) in AF is arrhythmogenic, affecting ion currents, the coupling of myocardial cells, and the extracellular matrix. It prolongs the potential for action, induces triggered activity, delays cardiac conduction, reduces repolarization, interferes with cell joints, and activates inflammatory pathways. Various OS biomarkers, such as uric acid and gamma-glutamyl transferase enzyme, vitamin C and E levels, and plasma antioxidant status, have been associated with AF development, severity, and recurrence [64].

The electron transport chain plays a crucial role in atrial remodeling during AF pathogenesis. Patients with paroxysmal AF have higher levels of total oxidant state and DNA damage than healthy controls [27]. ROS, which are physiological products of human metabolism, can result in homeostasis deregulation, affecting systemic balance and local levels, such as in the heart. Other sources of ROS in the AF scenario include NADPH oxidase, xanthine oxidases, NOS disconnection, myeloperoxidase, and monoamine oxidases [28,64]. NADPH oxidase contributes significantly to atrial OS and is associated with hyperglycemia, hyperlipidemia, hypertension, increased plasma fatty acid levels, and increased Ang II levels. The primary mechanism of NADPH oxidase activation is the increased activity of Ras-related C3 botulinum toxin substrate 1 (Rac1), leading to fibrosis through the positive regulation of connective tissue growth factor expression [64]. Antioxidant defense mechanisms include glutathione, superoxide dismutase, and thioredoxins. However, glutathione levels in the atrial tissues of patients with AF are low, possibly because of the downregulation of type L calcium flow due to S-nitrosylation caused by the accumulation of calcium induced by atrial tachycardia [64].

OS covers both aspects of the pathophysiological alteration of AF, contributing to electrical and structural remodeling, often coinciding [27,29]. A decrease in the action potential in the auricles and an increase in the heart rate [30] are observed, causing electrophysiological changes in K^+^ and Ca^2+^ currents and premature secondary depolarization. This leads to electrical heterogeneity, which is a crucial factor in promoting AF [27,31]. Oscillations in cardiac contraction are influenced by the levels of ROS and ions such as Ca^2+^, which, in excess, can result in the pathological opening of the mitochondrial permeability transition pore, superconnection in the excitation-contraction process, and maintenance of AF [27,31]. This is because Ca^2+^ has the ability to enter myocytes through voltage-dependent L-type calcium channels and activate ryanodine receptors (RyR2) in the sarcoplasmic reticulum, causing an increase in the release of Ca^2+^ ions [31]. Conversely, the oxidation of RyR2, driven mainly by mitochondrial OS, results in calcium leakage from the SR, constituting a possible therapeutic target for AF [65].

Ang II plays a role in electrical remodeling and shortening of the action potential and refractive period [32]. Activation of the renin-angiotensin system (RAS) is linked to AF, particularly in hypertension. This leads to fibrotic changes in the atrium, causing electrophysiological abnormalities and increasing the likelihood of AF development. Ang II stimulates the transformation of atrial fibroblasts into myofibroblasts, impeding AF remodeling. RAS inhibitors have been shown to reduce AF incidence in patients with hypertension [66].

Structural remodeling is linked to inflammatory processes, fibrosis, and aging and contributes to the persistence of arrhythmia [33]. Oxidative damage, particularly in patients with permanent AF, is mediated by hydroxyl and peroxynitrite radicals. These changes affect atrial myocyte energy generation and contractility, leading to significant changes in cardiac conduction parameters such as myocyte geometry, interstitial space size, and GAP joint conductivity and location [33].

ROS are crucial for cellular signaling and gene expression regulation, and their production and elimination imbalances can lead to mitochondrial damage, cell apoptosis, and cardiovascular diseases [34,35,64]. NADPH oxidase (NOX) is a critical enzyme in generating OS during AF [34], as it generates ROS, such as superoxide, triggering a cascade of reactions and changes, as illustrated in Figure 3 [36]. NOX has multiple isoforms that can influence various processes and contribute to chronic diseases, such as hypertension, hyperlipidemia, and HF [37]. Cardiovascular-related isoforms, NOX2 and NOX4, inhibit atrial remodeling and reduce inflammation associated with AF [37,38]. NOX activity is exacerbated in fibrillating atria, especially in the presence of systemic hormones such as Ang II and aldosterone [36]. Patients with permanent AF and paroxysmic AF showed a significant increase in NOX4 levels compared with non-AF subjects. Therapeutic approaches involving mitochondria, antioxidant use, and selective NADPH oxidase inhibitors offer substantial efficacy in treating AF [27].

Promising strategies focus on early interventions targeting the initial stages of ROS generation. However, the limited efficacy of antioxidant interventions when applied after oxidative damage has been established. Therefore, identifying OS in the early stages can contribute to the progression of chronic AF, allowing timely preventive interventions such as lifestyle modifications and vitamin supplements as antioxidant treatments [63].

Various molecules and medications have been investigated as potential therapeutic agents for AF. In this context, dipeptidyl peptidase-4 (DPP-4) inhibitors, selective sodium-glucose cotransporter 2 inhibitors (SGLT2-i), ubiquinone (coenzyme Q10, CoQ10), trimetazidine, and ranolazine, as well as experimental treatments targeting mitochondria and other biomolecular targets, such as relaxin-2, Costunolide, Febuxostat, and Wenxin Keli (WXKL), With these treatments, clinical benefits such as improved mitochondrial function, reduced postoperative AF, lower risk of new AF episodes, reduced mortality, decreased risk of AF recurrence after cardioversion, and reduced atrial electrical and structural remodeling are expected [27].

The cardioprotective effects of DPP-4 inhibitors are related to the mitigation of OS through the reduction of ROS, improvement of mitochondrial function, preservation of mitochondrial biogenesis, and reduction of inflammation. The therapeutic potential of DPP4 inhibitors was confirmed in an observational study by Chang et al. involving more than 90,000 patients with diabetes, in which the addition of a DPP-4 inhibitor as a second-line antidiabetic treatment reduced the onset of AF by 35% [27].

SGLT2-i has been shown to reduce arterial resistance by improving endothelial function, normalizing sodium and calcium cytosolic concentrations, reducing ROS synthesis, reducing systemic inflammation, and inhibiting atrial fibrosis and myocyte hypertrophy. They are more effective in reducing the risk of HF than DPP-4 inhibitors in patients with diabetes [27]. SGLT2-i can activate AMP-activated protein kinase (AMPK) in various tissues, suppress pro-inflammatory molecules, increase adiponectin levels, and reduce inflammatory markers in the myocardium [39]. Treatment with SGLT2-i has been shown to reduce mortality and hospitalization in patients with HF, regardless of the presence of diabetes. They also neutralize ROS production in cardiomyocytes, promoting alterations in atrial remodeling and reducing the AF load. However, the effects of SGLT2-i on the Ca^2+^ cycle, Na^+^ balance, inflammatory signaling, mitochondrial function, and energy balance are not yet conclusive [39]. A recent analysis of the DECLARE-TIMI 58 study showed a 19% reduction in the incidence of AF in patients with diabetes, regardless of pre-existing AF or HF. Effective AF prevention can transform this approach into HF treatment [39,40,41].

Metformin, a first-line antidiabetic drug, effectively prevents HF by mitigating atrial remodeling. A 13-year study of 645,710 patients with type 2 diabetes reported that metformin reduced the incidence of AF by 19%. It activates AMPK Src kinase and normalizes the expression of connectives, thereby decreasing the refractive period, induction, and duration of AF. Metformin also prevents atrial electrical and structural remodeling by activating the AMPK/peroxisome proliferator-activated receptor (PPAR)-γ coactivator 1α(PGC-1)/PPAR pathway and normalizing metabolic activity.

Metformin is concentrated in the mitochondria and helps preserve mitochondrial function by improving oxygen consumption and the activity of complexes I, II, and IV. It also promotes heart function by promoting mitochondrial respiration and biogenesis by upregulating PGC-1. This essential mitochondrial cofactor transports electrons from complex I to II and from complex II to III of the respiratory chain. CoQ10 is an effective antioxidant, membrane stabilizer, cofactor of mitochondrial disconnecting proteins, calcium-dependent channel stabilizer, metabolic regulator, and indirect regulator of cell growth and signaling molecule formation [29].

Exogenous CoQ10 supplementation can help treat cardiovascular diseases, including HF, AF, and myocardial infarction, as well as risk factors such as hypertension, insulin resistance, dyslipidemia, and obesity. In randomized clinical trials, inflammatory and OS indicators were significantly reduced in these diseases [68].

Trimetazidine, an anti-aging drug approved for ischemic cardiomyopathy, acts directly on the activity of the respiratory chain via the activation of complex I and normalizes the expression of regulatory factors of mitochondrial biogenesis. Although its beneficial action on mitochondrial function outside of the ischemic context has not yet been proven, its antiarrhythmic activity has been postulated to prevent structural atrial remodeling, reduce induction, and shorten the induction of AF [27].

Relaxin-2, a pleiotropic hormone, has significant therapeutic potential for treating AF. Elevated levels of relaxin-2 are associated with reduced expression of inflammatory markers, hydrogen peroxide concentration, inflammation, and OS genes. In vitro, treatment with relaxin-2 has demonstrated its ability to inhibit the migration of atrial heart fibroblasts and reduce the expression of profibrotic molecules [42].

Costunolide, a sesquiterpene lactone with anti-inflammatory and anti-fibrotic properties, reduces inflammation and fibrosis caused by Ang II in mice. Costunolide has been shown to preserve mitochondrial function and reduce OS, which are crucial for mitochondrial dysfunction [43]. Xu et al. [44] explored the effects of Febuxostat, an XO inhibitor, on AF susceptibility. They hypothesized that XO inhibitors could mitigate vulnerability to hypertension-related AF by improving the intracellular ROS environment and inhibiting the ox-Ca^2+^-calmodulin-dependent protein-kinase type-II (CaMKII) signaling pathway, which regulates heart contraction [44,45].

Both Febuxostat and Allopurinol significantly suppressed atrial remodeling related to hypertension and the perpetuation of AF. CaMKII oxidation and RyR2 hyperphosphorylation were restored, representing a breakthrough in our understanding of AF pathogenesis. Febuxostat also exerts its antioxidant effects by directly combating ROS. However, further clinical research is required to validate its use in the treatment of AF [44,45].

WXKL, a traditional Chinese medicine, treats various heart arrhythmias, including AF. A 2020 study by Gong et al. suggested that WXKL is essential for improving mitochondrial function, reducing OS, and preventing atrial remodeling in diabetic rats. This study showed that WXKL improves mitochondrial function, promotes increased basal and maximum mitochondrial respiration, and reduces endoplasmic reticulum oxidoreductase production. Its atrial selectivity in blocking the peak sodium stream is an essential feature of WXKL. It effectively regulates the activation of signaling pathways induced by hydrogen peroxide, preventing profibrotic cellular activity and thereby preventing atrial remodeling [46].

Andrographolide, an active ingredient in the medicinal plant *Andrographis paniculata*, has numerous pharmacological properties, including anti-hyperglycemic, antipyretic, anti-inflammatory, anticancer, anti-leishmaniosis, increased fertility, human immunodeficiency virus activity, cardiovascular benefits, immunomodulation, and choleretic action. Andrographolide has proven beneficial in AF by reducing heart cell apoptosis, improving mitochondrial function, demonstrating antioxidant properties, and regulating inflammation and calcium homeostasis genes. It also activates the transcription pathways involved in the antioxidant response, such as factor-2-related erythroid nuclear [47].

Elamipretide, also known as Bendavia, MTP-131, or SS-31, is a pioneering class of drugs that explicitly targets the mitochondria. It improves mitochondrial efficiency and reduces the production of ROS by stabilizing the mitochondrial membrane and cytochrome C, increasing ATP production, normalizing the ATP/ADP ratio, and reducing tumor nuclear factor (TNF) and C-reactive protein (PCR) levels. Other drugs targeting the mitochondria are under evaluation for their safety and effectiveness, and their potential to support mitochondrial function in AF prevention must be investigated from future perspectives [27].

Fibrates, which are PPAR agonists, are commonly used to treat hypertriglyceridemia, reduce hepatic apoC-III levels, and stimulate lipoprotein lipase-mediated lipolysis. They influence mitochondrial function via the PPAR/PGC-1 pathway. In animal experimental AF models, fenofibrate mitigates metabolic remodeling by regulating the PPAR-/sirtuin route 1/PGC-1, thereby reversing the shortening of the atrial refractory period. Bezafibrate positively affects mitochondrial biogenesis by increasing gene expression and mitochondrial DNA [27].

Therefore, an antioxidant-rich diet is essential for therapy [28]. Like vitamin E, vitamin C eliminates several ROS, such as O_2_, OH, peroxynitrite, sulfhydryl radicals, and oxidized low-density lipoprotein [29,34].

Other molecules have also been the target of the study, as seen in Table 2.

#### 3.1.2. Electrical and Arrhythmogenic Array

In this context, a disorder in intracellular calcium homeostasis constitutes a factor that retroactively drives mitochondrial dysfunction and, consequently, electrical remodeling. This is because the ion is involved in calcium-dependent mitochondrial processes. Calcium entering the cell triggers the subsequent release of calcium from the SR. It then binds to troponin C in fine filaments and is propelled by ATP, resulting in muscle contraction. The production of ATP, in turn, occurs through oxidative phosphorylation, a process also dependent on calcium, which involves the absorption of this ion by its corresponding mitochondrial uniporter. This capture triggers the activation of the tricarboxylic acid cycle and the movement of electrons through the (I–V) complexes of the electron transport chain. Therefore, calcium plays several crucial roles in mitochondria, including activating enzymes related to the Krebs cycle, regulating ATP production, and modulating the activity of the mitochondrial complexes responsible for the electron transport chain. Therefore, the inadequate management of intracellular calcium levels can result in mitochondrial dysfunction, which affects energy production and cardiac function. A precise balance of calcium concentration is essential for coordinating intracellular mitochondrial events and electrical processes for effective cardiac contraction.

Thus, activation of the renin-angiotensin-aldosterone system (RAAS) is involved in the pathophysiology of atrial remodeling in AF [33,48]. As mentioned by Hadi et al., the signaling pathway Ang II/Rac1/signal transducer and activator of transcription 3 (STAT3) is crucial in the atrial myocardium and participates in the structural remodeling of the atria [49]. In cultivated atrial myocytes and fibroblasts, Ang II-induced phosphorylation of tyrosine 3 transcription factors STAT3 using a Rac1-dependent mechanism was inhibited by Rac1-negative, losartan, and simvastatin. In atrial myocytes, activation of STAT3 by Rac1 involves a direct interaction between the two. An indirect paracrine effect on atrial fibroblasts mediated this activation. STAT3 activation, when constitutively active, resulted in increased protein synthesis, whereas the negative dominant form of STAT3 annulled Ang II-induced protein synthesis in atrial myocytes and fibroblasts. In addition, high levels of Ang II and phosphorus-STAT3 were detected in the atrial tissues of patients with AF [49].

Korantzopoulos et al. showed that the atrial tissues of patients with AF exhibit high levels of angiotensin-converting enzyme and increased Ang II receptor expression. Ang II and inflammation can increase the synthesis of superoxide, a significant cellular oxidative species, through the enzymes XO and NADPH oxidase [48,66]. Ang II stimulates the production of O_2_ through NADPH oxidase, activating the angiotensin type 1 receptor, and the inhibition of Ang II production contributes to the reduction of OS in vascular structures. Additionally, the atrial tissues of patients with AF show high angiotensin-converting enzyme levels and increased Ang II receptors [48]. As for aldosterone levels, patients with AF have high concentrations of this hormone, which are significantly reduced after the restoration of sinus rhythm [29].

The RAAS has received increasing attention owing to its role in developing heart conditions, including AF. Recent studies, such as that conducted by Zhao et al. (2020), have highlighted the beneficial effects of aliskiren (ALS) in mitigating atrial remodeling and reducing susceptibility to AF. Unlike other drugs that act on the RAAS, aliskiren directly inhibits renin. It binds to the active sites of renin, preventing the formation of Ang I and, consequently, RAAS activation. Previous studies have demonstrated that ALS suppresses changes in ion channel expression in chronic atrial tachycardia-induced AF models. However, Zhao et al. deepened their knowledge of the therapeutic potential of ALS in AF, noting its ability to mitigate atrial remodeling. ALS has been shown to reduce inflammation and OS and regulate the phosphoinositide 3-kinase (PI3K)/protein kinase B (Akt) signaling pathway, which plays an essential role in protecting heart tissue [50].

Ranolazine is an anti-anginal drug with antiarrhythmic properties that effectively inhibits sodium currents [27,34]. It reduces the likelihood of developing AF by approximately 50%, increases the success rate of cardioversion with amiodarone, and decreases the time required for sinus rhythm restoration. In addition, ranolazine improves mitochondrial function, mitigates OS, and suppresses apoptosis [27,34].

#### 3.1.3. Structural Rearrangement and Myocardial Fibrosis

Structural remodeling correlates with inflammatory processes, fibrosis, and aging, constituting the primary underlying mechanism for perpetuating this arrhythmia [33]. In these scenarios, cellular damage is usually mediated by hydroxyl and peroxynitrite radicals, which trigger considerable oxidative damage, particularly in patients with permanent AF [29]. These oxidative modifications have crucial effects on atrial myocyte energy generation and contractility [29]. From this perspective, it is evident that the three primary parameters undergo substantial changes, resulting in maladjustments in cardiac conduction: remodeling of the geometry of cardiac myocytes, modification of the size of the interstitial space, and variations in the conductivity and location of GAP joints [33].

Heart fibrosis is associated with various cardiovascular diseases and aging. The natural hypertrophy of cardiomyocytes in the context of fibrosis results from mechanical stress, such as stretching of muscle fibers. However, fibrosis disrupts the connection between muscle fibers and impairs electrical conductivity, impacting the conductance in the heart tissue and thus contributing to AF, as illustrated in Figure 4 [66]. Fibrosis plays a central role in structural remodeling. Hadi et al. highlighted alterations in the expression of nine genes that indicate the development of fibrosis in patients with AF. Several fundamental signaling routes have been proposed based on experiments using animal models and human studies. OS, inflammatory processes, and Ang II and transformative growth factor beta (TGF-β) influence fibrosis associated with AF. These factors, among others, regulate intermediates in the signaling pathways, such as NADPH oxidase, MAP kinases, and nuclear factor kappa B (NF-κB). In addition, TNF-α expression is increased in patients with AF. A comparison of right appendicular atrial samples among patients with and without AF revealed a significant increase in right appendicular atrial fibrosis and TNF-α protein expression in patients with AF [71].

Increased activity of Rac1, driven by Ang II and TGF-β1, seems to be the main mechanism of activation of NADPH oxidase in this context, resulting in fibrosis through increased expression of connective tissue growth factor. In addition, the enzyme myeloperoxidase, released by activated polymorphonuclear neutrophils, is associated with atrial fibrosis and remodeling [63]. A recent study by Yagi et al. showed that pitavastatin reduces the incidence of Ang II-induced atrial fibrillation, increases the left atrium, mitigates fibrosis and heart hypertrophy, and negatively regulates Rac1 activity in mice [72].

Galectin-3 (Gal-3) belongs to the lectin family and plays a role in cell differentiation, fibrinogenesis, and inflammation. Gal-3 induces fibrosis by activating fibroblasts and endocardial cells, thereby increasing the extracellular matrix. Evidence indicates that Gal-3 plays an essential role in the acute phase of the inflammatory response, triggering the activation of neutrophils and mast cells, and is involved in the transition to chronic inflammation, resulting in fibrogenesis and tissue fibrosis. In addition, the N-terminal prohormone of brain natriuretic peptide is correlated with the degree of fibrosis in the atria of patients with AF and serum collagen remodeling markers [73].

Hydrogen sulfide (H_2_S) has been studied in this context because of its beneficial properties in human physiology and potential role as a cardiovascular protector. Studies have suggested that H_2_S plays a role in reducing AF and mitigating atrial fibrosis. These beneficial effects are related to the PI3K/Akt/eNOS signaling pathway, which regulates the production of ROS and, therefore, participates in maintaining the redox balance in the heart tissue. H_2_S can significantly increase the activation of this signaling pathway, which leads to a reduction in atrial fibrosis. Furthermore, H_2_S does not seem to negatively affect glucose metabolism, which is relevant in the context of DM. H_2_S can, therefore, be a potential and promising strategy for mitigating atrial fibrosis and reducing the incidence of AF in patients with diabetes without exacerbating the metabolic imbalances associated with DM [51].

Thiazolidinediones prevent AF and reduce its recurrence after electrical cardioversion [27]. Pioglitazone is responsible for prolonging glycemic control because of its ability to increase beta-cell activity and reduce insulin resistance. The drug has beneficial effects on cardiac risk factors, substitutes for indicators of cardiovascular disease, and reduces the frequency of cardiac events in individuals with diabetes. Individuals with non-diabetic insulin resistance have a lower risk of recurrence of transitory ischemic attacks and ischemic strokes. Pioglitazone reduces the risk of AF by inhibiting atrial remodeling. In addition, it is effective in treating the symptoms of diseases associated with insulin resistance, such as nonalcoholic steatohepatitis and polycystic ovary syndrome. Pioglitazone is an effective drug for patients with insulin resistance and diabetes; however, it is underused because of the toxicity of other thiazolidinediones [74].

### 3.2. OS Modulators

#### 3.2.1. Inflammation

OS and inflammation are intrinsically related, and several clinical studies have shown a strong association of inflammatory pathways with the presence and recurrence of early or late AF [29,49,52]. Local and systemic OS proceeds with producing and releasing pro-inflammatory cytokines, such as interleukin (IL)-1β, 2, 6, 8, 10, and 12, C-reactive protein (CRP), vascular endothelial growth factor, TGF, TNF-α, and CD40. The OS and inflammatory state generate changes in intracellular calcium management and the concentrations of this ion, resulting in electrical remodeling, shortened heart cycles, and recurring episodes of AF [49].

Inflammasomes play a crucial role in the innate immune response and the maintenance of tissue homeostasis. The NOD-, LRR-, and pyrin domain-containing protein 3 (NLRP3) inflammasome activates caspase-1 and mature ILs 1 and 18, which play significant roles in the cardiovascular system. In addition, high-cholesterol diets in the Western world can trigger inflammatory responses dependent on the NLRP3 inflammasome, thus influencing the reprogramming of the innate immune system. The inflammatory process involves the production of various cytokines, primarily generated by the mitochondria, through different molecular pathways associated with the elevation of ROS levels. This triggers the activation of the inflammasome NLRP3. The influence of acute and chronic inflammation, along with OS, has been associated with the triggering of atrial and ventricular fibrillation, as evidenced by experimental and clinical studies. Cardiovascular risk factors, such as hypertension, obesity, insulin resistance, metabolic syndrome, aging, and neurological disorders, cause a state of inflammation and OS, predisposing patients to heart arrhythmias. Disturbances in the redox state and an increase in pro-inflammatory cytokines can cause cardiac channelopathy, affecting ion channels and gap junction canals, such as connexin (Cx)43 and Cx40, as well as the activation of connexin hemichannels and abnormal calcium regulation. These changes are crucial in the induction and maintenance of atrial and ventricular fibrillation. Dysbiosis of the intestinal microbiota, which produces bioactive metabolites, can contribute to pro-arrhythmic, inflammation-related actions. Environmental factors such as air pollutants and artificial night lighting can also predispose patients to cardiovascular disorders related to inflammation and OS, thereby promoting heart arrhythmias. Therefore, connexin-hemichannel inhibition may be an effective strategy for treating inflammation and preventing arrhythmias. Furthermore, recent studies have revealed the inflammatory functions of heart immune cells, highlighting that leukocytes can be arrhythmogenic, affect tissue composition, or interact with cardiomyocytes. Recent studies have also identified the presence of genetically determined systemic inflammation in cardiovascular diseases, highlighting the NLRP3 inflammasome as a promising therapeutic target (Table 3) [53].

Furthermore, NF-κB is crucial in regulating gene transcription in response to the redox state. This is relevant to conditions of injury and inflammatory stress. NF-κB negatively regulates the transcription of the heart’s sodium channel in response to OS, suggesting that it may influence other aspects of the pathophysiology of AF and serve as a therapeutic target [71]. In addition, its activation results in the expression of genes involved in inflammation, such as TNF-α, iNOS, IL-1β, and matrix metalloproteinases. These processes contribute to activating oxidases such as NOX, triggered by agents such as Ang II and atrial stretching, creating a vicious cycle in which the activation of NOX promotes AF and the AF itself contributes to NOX activation [72].

In the context of the evaluation of inflammatory markers, CRP [6] has been highlighted. It plays a role at the systemic level, with pro-inflammatory and anti-inflammatory actions [49], and locally, it influences nitric oxide deficiency, contributing to increased thrombogenic risk [54,55]. CRP levels may be elevated in some patients with AF; however, the underlying mechanism has not yet been fully clarified [49]. In addition, in cases of permanent AF, we expected to find more significant levels of high-sensitivity C-reactive protein (hs-CRP) than in paroxysmal AF [54]. As the disease progresses and becomes chronic, structural remodeling of the left atrium occurs, often indirectly increasing hs-CRP values [49]. Ang also plays a significant role in inflammatory processes.

The increase in plasma levels of IL-6, PCR, and plasma viscosity corroborates the existence of an inflammatory state in patients with chronic AF. These inflammatory indices are related to the prothrombotic state. They may be linked to the clinical condition of patients with underlying vascular diseases and comorbidities, not just the presence of AF [49]. However, AF provides a state of hypercoagulability even in the absence of an underlying heart disease [48,49]. Abnormalities in hemostasis, fibrinolysis, endothelium, and platelets may be present in AF, which may increase the risk of thromboembolism and stroke. An explanation for the high incidence of thromboembolism in patients with AF, with or without valvular heart disease, is the high levels of beta-thromboglobulin and platelet factor 4 [49]. Furthermore, Ferro et al. found that increased soluble CD40L levels predicted vascular events in patients with AF. This reinforces the hypothesis that increased platelet activation affects clinical progression [49].

PCR positively correlated with the risk of stroke and was related to risk and prognostic factors, such as mortality and vascular events. High IL-6 levels are independent predictors of stroke or death in patients at high risk of AF. In addition, high levels of F1.2 are associated with clinical risk factors for stroke in AF, while increased levels of beta-thromboglobulin are linked to manifestations of atherosclerosis [49].

In patients with AF, increased plasma levels of sCD40L were observed, which correlated with increased levels of vascular endothelial growth factor, angiopoietin-2, and tissue factor. This interaction among platelets, angiogenic markers, and tissue factors may play a role in determining the origin of the prothrombotic state associated with AF. In addition to sCD40L, patients with AF show significantly higher levels of monocyte-1 chemotactic protein (MCP-1), hs-CRP, intercellular adhesion molecule (ICAM), and vascular cell adherence protein (VCAM), which reach higher levels in patients with atrial thrombosis. VCAM and MCP-1 are independent predictors of atrial thrombosis and ischemic stroke in patients with AF. However, even after successful direct electrical cardioversion, the recovery of ICAM, VCAM, and MCP-1 levels and CD40 expression in platelets can take up to five weeks because of the persistence of underlying atrial structural abnormalities, which are not normalized immediately with the restoration of sinus rhythm [49,55].

AF is not an absolute prerequisite for developing prothrombogenic changes in the atrial endocardium. In many patients with AF, arrhythmia may be secondary to pre-existing structural changes that are prothrombogenic in the atrial myocardium and endocardium, a phenomenon known as “endocardial remodeling” [49].

In addition, Hadi et al. revealed that platelet P-selectin levels were considerably lower in patients with AF who did not receive antithrombotic therapy than in healthy patients. Even if P-selectin is expressed on the platelet surface in AF, the absolute levels of plateletvP-selectin are low. This result can be explained by two theories: (1) P-selectin is reduced in the platelets after platelet activation, consequently increasing in the plasma. (2) P-selectin, expressed on the platelet surface during activation, changes its configuration and can be detected using a specific antibody during flow cytometry. Since this study used an enzyme-linked immunosorbent assay test to detect platelet P-selectin, the antibody may have detected only P-selectin granules without recognizing membrane P-selectin. The second hypothesis is supported by the fact that membrane P-selectin represents 90% of the P-selectin in the lysate, whereas 10% is sP-selectin in the membrane of platelet granules [49].

Concerning thrombotic events, CRP was positively correlated with the risk of stroke, related to risk factors and prognosis, and associated with the risk of AF recurrence. Patients with a moderate-to-high risk of stroke have lower levels of CD40 ligands [49] and higher levels of CRP [29,49]. Simvastatin effectively modulates CD40 expression and may contribute positively to reducing the risk of intra-atrial thrombosis [49]. In addition, CRP-reducing therapies with statins, such as atorvastatin, can prevent AF and electrical and structural remodeling by preventing inflammation [29,34,49]. Statins inhibit OS by preventing the synthesis of free oxygen radicals induced by NADPH oxidase [56].

Neuman et al. reinforced the hypotheses that statins prevent electrical remodeling in rapid stimulation-induced AF, reduce the load of AF after surgery, and prevent the recurrence of AF following cardioversion [56]. In addition, statins induce Kruppel-like transcription factor 2 in endothelial cells, an essential mediator of cell quiescence that regulates the expression of several target genes and promotes an anti-inflammatory and antithrombotic endothelial phenotype [19].

Statins also appear to mitigate the signaling of growth factors activated by Ang II, thrombin, endothelial growth factor, platelet-derived growth factor, and profibrotic growth factor β; they can exert a potent anti-inflammatory and antithrombotic effect on blood monocyte macrophages, reducing the expression of pro-inflammatory cytokines and coagulation factors; they can indirectly reduce the cytotoxic activity of T cells against endothelial cells; and they can activate the pathway of phospholipase A-cyclooxygenase, shifting the balance to a greater synthesis of prostacyclin, a vasodilator and an anti-inflammatory [19].

Corticosteroids are anti-inflammatory drugs with immunomodulatory properties. This class of drugs appears to have indirect antioxidant properties, mainly due to the attenuation of the inflammatory state. There is scientific evidence that corticosteroids significantly reduce CRP and atrial endothelial NOS levels.

Carvedilol is a slightly selective β1 blocker that becomes non-selective at higher doses. It has α1 blocking and antioxidant properties, modulating effects on various ion channels and currents. In addition, it is superior to other selective β1 blockers, such as metoprolol and atenolol, in suppressing postoperative AF. The hypothesis that carvedilol is superior to other beta-blockers in treating AF is explained, at least in part, by its antioxidant effects [29].

The anti-inflammatory effects of omega-3 fatty acids are remarkable and can be attributed to substituting arachidonic acid in cell membranes. Unlike arachidonic acid, which is a precursor of pro-inflammatory mediators such as prostaglandins and thromboxane, omega-3 fatty acids, such as eicosapentaenoic acid and docosahexaenoic acid, promote the formation of anti-inflammatory mediators similar to resolvins and proteins that inhibit pro-inflammatory cytokines. The modulation of ionic and conveyor channels, as well as the properties of the cell membrane, is one of the direct actions of omega-3 fatty acids that can influence the occurrence of AF. Supplementation with omega-3 fatty acids affects the function of ion channels in generating cardiac action potentials, stabilizing electrical activity, and prolonging the refractory period of cardiomyocytes. In addition, omega-3 fatty acids can preserve the heart’s structural integrity, partly through the modulation of proteins such as Cx43, which is essential for the function of cardiac GAP joints and, therefore, for preventing structural remodeling of the heart. Another crucial aspect of the action of omega-3 fatty acids is their ability to reduce OS in heart cells. However, the effectiveness of omega-3 fatty acids in preventing AF may depend on the clinical background and individual conditions. Clinical studies have shown varying results, and more research is needed to determine the ideal circumstances under which supplementation with omega-3 fatty acids can be the most effective [53], as shown in Table 3.

#### 3.2.2. Genetics

ROS plays a role in gene regulation and contributes to the induction and maintenance of AF through various mechanisms. This includes interactions with proteins, nucleic acids, and other molecules that can alter the structure of the atrium and cause tissue damage [35]. Furthermore, specific genes are positively and negatively regulated in response to the clinical conditions of AF. The positive regulation is associated with monoamine B oxidase, which increases the release of hydrogen peroxide and calcium ions. In contrast, negative regulation is conducted by enzymes such as glutathione peroxidase [35]. Adenosine, produced by the degradation of ATP and ADP in cardiomyocytes and endothelial cells, exerts cardioprotective effects by activating adenosine receptors [57]. It has been shown that patients with AF have increased expression of the adenosine A2A receptor in the right atrium compared with non-AF subjects, which suggests the contribution of this receptor to the development of AF [75].

In most cases of AF, predisposing factors include systemic and cardiac disorders such as hypertension, HF, and valvular diseases. These conditions eventually lead to atrial enlargement, fibrosis, and electrical abnormalities. In less common situations, AF may be primarily triggered by an isolated electrical disorder or a genetic predisposition, as evidenced by recent large-scale genomic association studies. This is due to the relatively rare mutations in cardiac potassium and sodium channels and RyR2 receptors [58]. Several experimental studies have indicated that AF is associated with modifications in genetic regulation, likely contributing to the positive feedback cycle perpetuating this arrhythmia. Studies conducted on families with AF predisposition have identified several gene loci that significantly regulate susceptibility to this condition, including 11p15, 21q22, 17q, 7q35–36, 5p13, 6q14–16, and 10q22. Some of these loci encode subunits of cardiac potassium channels, such as KCNQ1, KCNE2, KCNJ2, and KCNH2. In addition, other studies have identified genes related to potassium and sodium channels (SCN5A), structural proteins such as sarcolipin, regulators of the RAS, genes that influence the coupling between cells, and genes related to OS and inflammatory mediators. In this context, potassium channels are essential for determining the resting membrane potential and cell repolarization after the occurrence of an action potential, and changes in the activity of these canals, whether an increase or decrease, may predict a greater susceptibility to AF. Significant reductions were also observed in the expression of mRNA encoding the alpha subunits of L-type calcium channels. These observations indicate that transcriptional regulation is a molecular mechanism underlying the alterations in ion channel expression and atrial electrical remodeling. Gene expression profiles comparing individuals with AF and healthy individuals suggest that transcription factors are affected by AF and that some of these factors are involved in redox signaling [71].

Liu et al. investigated the association between mitochondrial transcription factor A (TAFM) and AF and its effect on cardiomyocyte ATP content. Left atrial appendix samples were collected from 20 patients with a normal sinus rhythm and 20 patients with AF. TAFM expression levels were evaluated in both tissues. A tachypacing model was constructed to assess ATP content, cell viability, and expression levels of TAFM, NADH-1 mitochondrial-coded dehydrogenase, cytochrome c oxidase-1 mitochondrially-coded, central subunit 1 of ubiquinone reduction-oxidation of NADG, and subunit 6C of cytochrome C oxidase. The effects of overexpression and inhibition of TAFM have also been investigated. The results showed that TAFM expression levels were reduced in tachypacing AF tissues and cardiomyocytes, and the restoration of TAFM increased ATP content through the positive regulation of NADH 1 mitochondrial-coded dehydrogenase and cytochrome c oxidase-coded expression levels in tachypacing cardiomyocytes. These findings suggest that TAFM may be a new therapeutic target for treating patients with AF [59].

Genetic therapy is a promising therapeutic approach; however, it remains experimental. Generally, genetic constructs inserted into adenoviral vectors are used to deliver genes directly to the heart muscle, either by direct injection into the muscle, application to the surface of the heart, or infusion through the coronary arteries. These gene-based approaches successfully restored the average heart rate and improved heart rate control in animal models of AF. However, they have yet to reach widespread clinical use, as shown in Figure 5 [27].

#### 3.2.3. Damage to Mitochondrial DNA

With the aging process and accumulation of exogenous stress, as in the case of AF, free radicals and ROS can overload the antioxidant system, damaging cellular structures such as lipids, proteins, and DNA [27,76]. Increased OS makes mitochondrial DNA (mtDNA) susceptible to damage and mutations owing to its inadequate and ineffective repair ability during replication. This leads to the progressive accumulation of damage to mtDNA, which is significantly higher in patients with AF than those with sinus rhythm [76]. The hypothesis for this finding is related to managing intracellular calcium levels. Owing to the high amount of Ca^2+^ in the cytoplasm, mitochondria absorb large amounts of this ion to maintain intracellular ion homeostasis. This accumulation of calcium in the mitochondrial matrix of heart cells in patients with AF can alter the potential of the mitochondrial membrane, promoting the reduction of ATP synthesis, excessive production of ROS, inhibition of antioxidant mechanisms, and damage to lipids, proteins, and DNA [29,48,76]. In parallel with the levels of oxidative lesions, patients with AF present cardiomyocytes with a higher content and mass of mtDNA, increased synthesis of mitochondrial respiratory enzymes, and increased proliferation of mitochondria, both in bodily tissues under OS and in the affected tissues of patients with mitochondrial myopathies [31,49]. This high number of mtDNA copies may result from a compensatory feedback mechanism owing to an impaired respiratory chain [76].

A study conducted by Lin et al. identified a series of accumulated oxidative lesions in the atrial muscle mtDNA of patients with AF, with the deletion of 4977 mtDNA base pairs being the most relevant and significantly found in patients with AF. This was accompanied by higher levels of 8-OHdG, a modified nucleic acid that is an essential indicator of oxidative DNA damage. In addition, the activity of mitochondrial complexes I and II decreases considerably, whereas that of complex V increases in patients with AF, accompanied by an increase in superoxide production. Owing to acute OS and excessive accumulation of Ca^2+^ in mitochondria, mtDNA is rapidly damaged, resulting in mitochondria with impaired bioenergetic functions, especially in mitochondrial complexes I and II. This generates excellent production of ROS in the atrial muscles of patients with AF. Despite a compensatory mechanism involving the activity of oligomycin-sensitive ATPase, the compensatory production in mitochondria is not practical for reducing OS. It accelerates the cycle of oxidative damage in various cellular components, with an emphasis on mtDNA. Although some mtDNA repair enzymes can mitigate damage, most defective mtDNA molecules can escape repair and degradation processes, resulting in the progressive accumulation of oxidative damage [27,31,49,76].

#### 3.2.4. Aging and Comorbidities

The systemic conditions of the human body—the presence of hypertension; HF; DM; and obesity—predispose patients to the most varied clinical situations and associated comorbidities; including arrhythmias such as AF [60,61]. Therefore, aging is one of the most relevant risk factors for the onset of AF, as it correlates with increased oxidative damage [29,33,35]. With increasing age, certain conditions can favor this clinical condition, such as DNA deletions, fibrosis, hypertrophy, and electrical and structural cardiac remodeling [31,33]. In addition, aged heart muscles usually experience calcium overload, which influences the process of electrical remodeling, as calcium has a central influence on cell-cell adherence [64]. Obesity, in turn, is a systemic condition that can induce the appearance of AF by direct mechanisms—infiltration of adipocytes in the atrial heart muscle—and indirect mechanisms—an increase in pro-inflammatory cytokines and a change in the characterization of macrophages from M2 to M1; favoring inflammation [32].

### 3.3. NF-KB

AF-associated tachycardia triggers mitochondrial dysfunction and OS, which induce pro-inflammatory pathways through inflammasome activation involving NF-KB, caspase-1, and NLRP-3 [27,48]. The activation of the NF-KB signaling pathway is accompanied by the induction of the expression of the target genes of NF-KB in the atrial tissue, as shown in Figure 6 [48,62].

Compared with patients with sinus heart rhythm, the atrial tissue of patients with AF shows a significant increase in carbonylated proteins, a decrease in the content of tissue-free thiols, and an increased nuclear presence of NF-κB. In addition, the expression of NF-κB target genes, such as lectin-like oxidized LDL receptor-1, ICAM-1, and heme oxygenase-1, is increased in patients with AF, with a more pronounced elevation during fibrillation. Heme oxygenase-1, a redox-sensitive induction protein, plays a role in cytoprotection against OS. Simultaneously, excess lectin-like oxidized LDL receptor-1 contributes to increased superoxide production and adhesion molecule expression, resulting in an ascending regulatory process [48].

In addition, an increase in NF-κB inhibitor alpha (IκBα) phosphorylation was observed, a crucial regulatory step in NF-κB activation. This phosphorylation directs IκkBα to polyubiquitination and subsequent degradation mediated by the proteasome. The release of IκkBα exposes NF-κBs nuclear location signal, facilitating its translocation. Intracellular increases in calcium and ROS trigger an immediate response of the transcription factor NF-κB. Additionally, OS mediates the prothrombotic response to target genes by activating NF-κB signaling [48].

## 4. Conclusions

Currently, the treatment of AF mainly involves anticoagulation and therapies to control the heart rate or rhythm. The choice between these therapeutic approaches for specific patient groups remains debatable. In addition, attention has been focused on modifiable risk factors, emphasizing underlying cardiovascular diseases, comorbidities, and their relationship with AF.

Therefore, understanding the biomolecular mechanisms underlying AF pathophysiology is paramount for advancing research and developing new therapeutic modalities. An in-depth study of these mechanisms provides a solid basis for identifying specific therapeutic targets that can be targeted more accurately and effectively to control symptoms and address the underlying causes of arrhythmia. A better understanding of the biomolecular mechanisms of AF will enable the exploration of new therapeutic approaches, including the development of targeted drugs, genetic therapies, and innovative treatments. In addition, it helps optimize existing interventions, making them more targeted and personalized to meet the needs of specific patients.

## Figures and Tables

**Figure 1 ijms-25-00535-f001:**
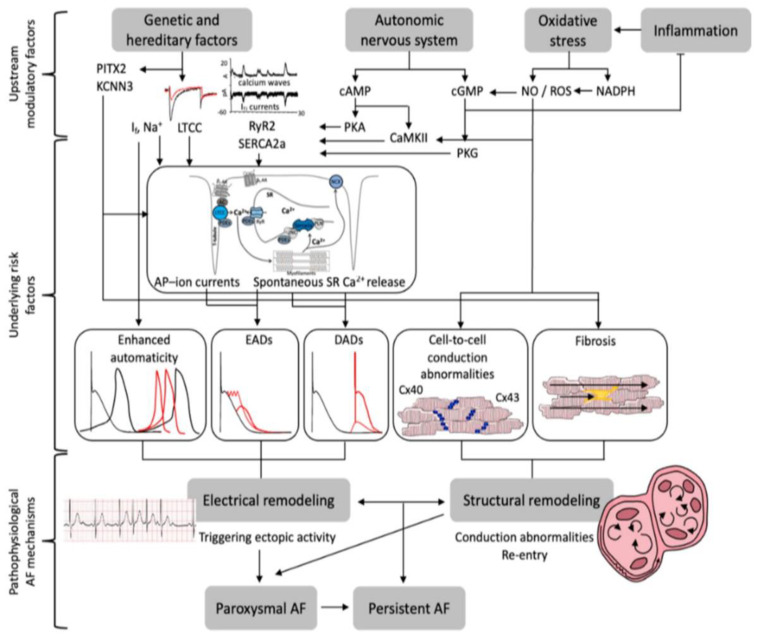
Pathophysiological mechanisms underlying the initiation and maintenance of atrial fibrillation (AF). Summary of upstream modulatory factors, underlying risk factors, and pathophysiological AF-associated mechanisms. Abbreviations: PITX2, paired-like homeodomain transcription factors 2; KCNN3 = Potassium Ca^2+^ctivated Channel; If = funny current; LTCC = L-type Ca^2+^ channel; RyR2 = ryanodine receptor 2; SERCA2a = SR− Ca^2+^ ATPase; cAMP, 3′,5′-cyclic adenosine monophosphate; PKA and PKG, protein kinase A and G; CaMKII = Ca^2+^-calmodulin dependent protein-kinase type-II; NO = nitric oxide; ROS = reactive oxygen species; NADPH =nicotinamide adenine dinucleotide phosphate; cGMP = cyclic guanosine monophosphate; AP = action potential; SR = sarcoplasmic reticulum; EADs/DADs = early/delayed afterdepolarizations; Cx40/43 = connexin 40/43; AF, atrial fibrillation. Source: [26].

**Figure 2 ijms-25-00535-f002:**
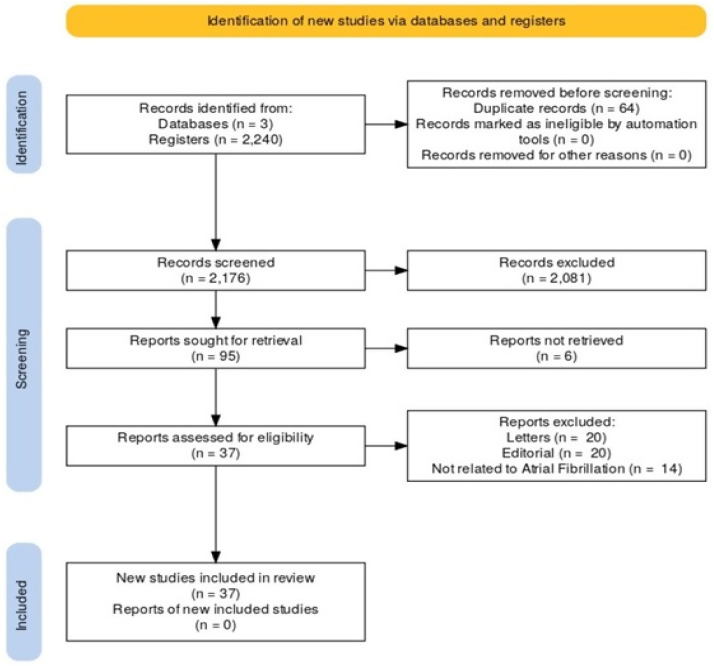
Preferred Reporting ltems for Systematic Reviews and Meta-Analysis flowchart demonstrating the identification, screening, and studies included in this research.

**Figure 3 ijms-25-00535-f003:**
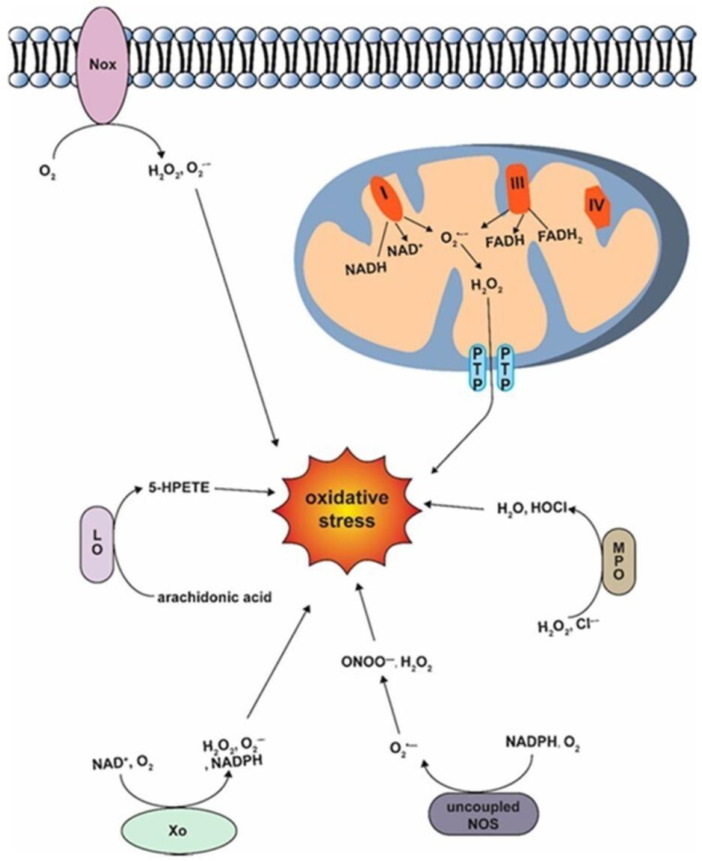
Diagrams of oxidative stress production pathways. Source: [67].

**Figure 4 ijms-25-00535-f004:**
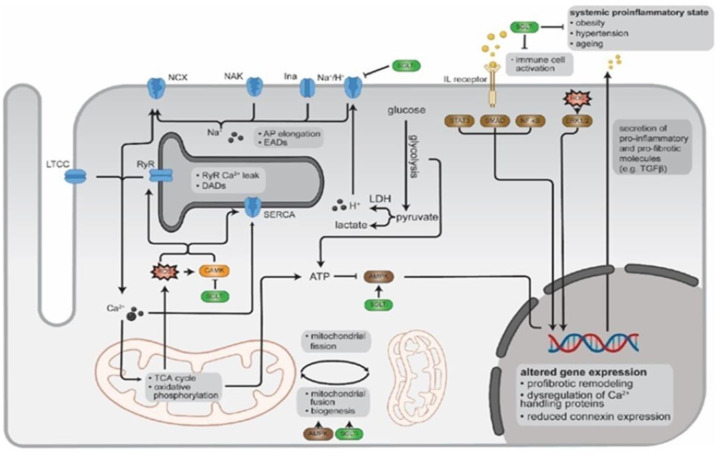
Molecular mechanisms of SGLTi on redox signaling in cardiomyocytes. Source: [39].

**Figure 5 ijms-25-00535-f005:**
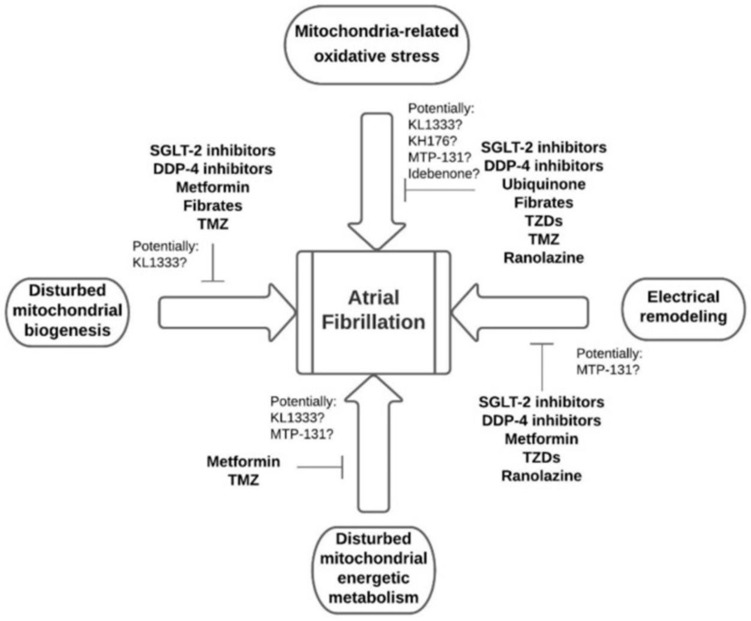
Pharmacological interventions target the basic mechanisms of atrial fibrillation. Abbreviations: SGLT2, sodium-glucose cotransporter 2; DDP-4, dipeptidyl peptidase-4; TZDs, thiazolidinediones; TMZ, trimetazidine. Source: [27].

**Figure 6 ijms-25-00535-f006:**
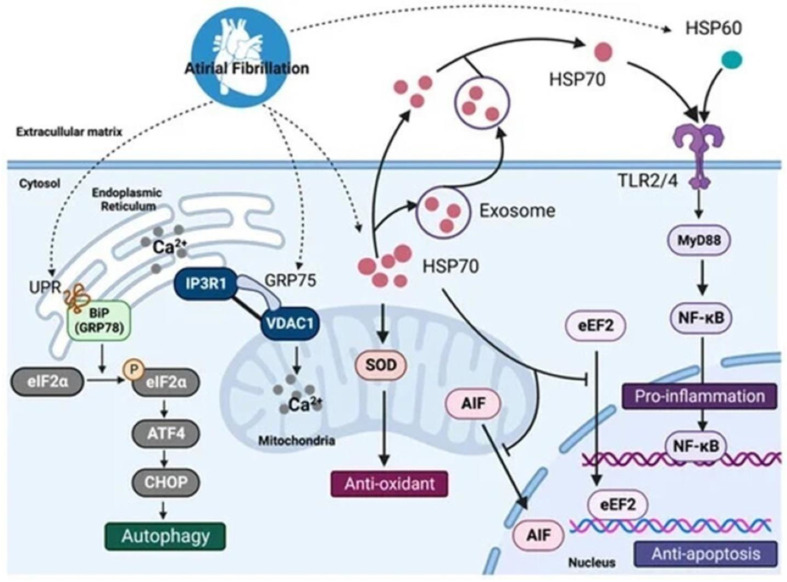
The mechanisms regulated by HSPs in atrial fibrillation. Intracellular HSP70s and HSP60 participate in autophagy, ER-stress, and oxidative stress reactions, exerting antioxidative and antiapoptotic effects and preventing atrial remodeling. However, when released into the extracellular matrix in a free state, HSP60 and HSP70 bind to TLRs on the surface of the membrane to induce inflammatory responses by activating the MyD88/NF-κB pathway. HSP, heat shock protein; UPR, unfolded protein response; GRP78, glucose-regulated protein 78; eIF2α, α-subunit of eukaryotic initiation factor 2; ATF4, activating transcription factors 4; CHOP, CCAAT/enhancer-binding protein homology protein; IP3R1, inositol 1,4,5-trisphosphate receptors; VDAC1, voltage-dependent anion channel-1; SOD, superoxide dismutase; AIF, apoptosis-inducing factor; eEF2, eukaryotic elongation factor 2, NF-κB, nuclear factor kappa B, TLR4, Toll-like receptors 4. Source: [62].

**Table 1 ijms-25-00535-t001:** Baseline Characteristics of the Reviewed Articles and the Studied Population.

Author/ Year	Country	Title	Type of Study	Study Population	Objectives	Methods	Main Findings
Emelyanova 2016 [15]	USA	Selective downregulation of mitochondrial electron transport chain activity and increased oxidative stress in human atrial fibrillation	Case-control	Right atrial tissue between non-AF and AF patients undergoing open-heart surgery.	The study compared mitochondrial oxidative phosphorylation (OXPHOS) complexes and oxidative stress.	Between 2012 and 2016, elective open-heart surgery patients were examined for atrial appendage tissues, examining factors like O_2_ oxidoreductase activity, mitochondrial OXPHOS complexes, citrate synthase activity, and protein expression.	AF is linked to decreased ETC activity and increased oxidative stress, which can contribute to the progression of the substrate for AF.
Muszyński. 2021 [27]	Switzerland	Mitochondrial dysfunction in atrial fibrillation—mechanisms and pharmacological interventions	Review	Atrial fibrillation	The study seeks to review mitochondrial dysfunction mechanisms in atrial fibrillation, exploring their role and potential therapeutic implications, aiming to enhance our understanding and contribute to improved prevention and treatment strategies for arrhythmia.	This text aims to comprehend the pathophysiological mechanism of arrhythmia, its prevention methods, and the role of mitochondrial dysfunction in the onset of arrhythmia.	Mitochondrial dysfunction, a key factor in cardiomyocyte integrity and cardiac performance, can be addressed through medications that target energetic imbalance, metabolic disturbance, and oxidative stress.
Ren, 2018 [28]	China	Mechanisms and treatments of oxidative stress in atrial fibrillation	Review	Atrial fibrillation	The study aims to review pathways of reactive oxygen species (ROS) induction in atrial fibrillation (AF), focusing on electrical and structural remodeling. It explores sources and factors contributing to AF-related oxidative stress, identifying potential therapeutic targets through scavenging oxidative stress markers.	The research uses atrial electrical and structural remodeling data to explore various reactive oxygen species pathways that can induce atrial fibrillation (AF).	AF-related oxidative stress, triggered by factors like NADPH oxidase activation, calcium overload, and cardiovascular conditions, may be a therapeutic target for AF.
Korantzo poulos. 2007 [29]	Greece	The role of oxidative stress in the pathogenesis and perpetuation of atrial fibrillation	Review Article	Atrial fibrillation	This summary critically analyzes the current literature on AF and oxidative stress, highlighting the scientific background of antioxidant therapeutic interventions.	The authors conducted a comprehensive search using the MEDLINE database, reference lists, and significant cardiovascular journal supplements to identify studies published up to December 2005.	Oxidative stress is linked to AF pathophysiology, contributing to arrhythmia perpetuation. Modulating oxidative stress can benefit AFs development of inflammatory and electrophysiological changes.
Schillinger, 2012 [30]	USA	Atrial fibrillation in the elderly: The potential contribution of reactive oxygen species	Review	Atrial fibrillation	The study explores the link between aging and AF through reactive oxygen species’ (ROS) oxidative damage, which induces intracellular and extracellular changes.	An overview of AF pathophysiology and introduces the critical structures that predispose an otherwise healthy atrium to AF when damaged. The available evidence that ROS can lead to damage to these vital structures is then reviewed. Finally, the evidence linking the aging process to the pathogenesis of AF is discussed.	To be certain, through serving as agents of oxidative cellular damage, ROS is likely to be a major causative factor in the development of AF. The cellular changes brought about by ROS-mediated damage are sufficient to promote tissue changes consistent with AF triggers and atrial electrical and structural remodeling.
Mason, 2020 [31]	Germany	Cellular and mitochondrial mechanisms of atrial fibrillation	Review	Atrial fibrillation	This review discusses the importance of mitochondrial Ca^2+^ handling in regulating ATP production and mitochondrial ROS emission and how alterations may play a role in AF, particularly in these aspects of mitochondrial activity.	It involves understanding the molecular mechanisms underlying AF and specific treatment options.	In the current review, we discussed the hypothesis that remodeling and increased energy demand during AF lead to oxidative stress, shifting the redox environment to a state of energy deficit and compromised ROS scavenging capacity.
Mascolo, 2020 [32]	Italy	Angiotensin II and angiotensin 1–7: which is their role in atrial fibrillation?	Review	Atrial fibrillation	This review aims to provide an overview of the evidence for the possible role of the two RAS pathways (classic and non-classic) in the pathophysiology of AF, the proposed cellular and molecular mechanisms, and the results of clinical studies with classic RAS antagonists.	Understand the role of the RAS in the induction of AF, influenced by inflammatory and cardiac processes, electrical remodeling, and epicardial fat accumulation.	In this review, we summarized the evidence showing that both RAS pathways can balance the onset of AF through different biological mechanisms involving inflammation, epicardial adipose tissue (EAT) accumulation, and cardiac remodeling.
Tribulova, 2015 [33]	Slovak Republic	New aspects of the pathogenesis of atrial fibrillation: Remodeling of intercalated discs	Review	Atrial fibrillation	Understanding the role of key factors of aging, oxidative stress, and inflammation in the development of age-related cardiovascular disease and AF.	Review of reactive oxygen species, their production, and relationship with systemic inflammation. Approach to studies that consider cardiac structural and electrophysiological remodeling as crucial for developing and maintaining atrial fibrillation.	Data suggest that alterations in atrial connexin-43 and/or connexin-40 expression, phosphorylation, and distribution affect cell-to-cell electrical coupling and molecular signaling that is proarrhythmogenic. Gap junctional connexin channels are considered targets for arrhythmia prevention, and therapeutic interventions for mitochondria-related reactive oxygen species appear. In addition, aging is accompanied by abnormalities in adhesive junctions that most likely promote asynchronous contractions and arrhythmias.
Pool, 2021 [34]	Switzer land	The role of mitochondrial dysfunction in atrial fibrillation: translation to druggable targets and biomarker discovery	Review	Atrial fibrillation	To dissect molecular mechanisms that drive AF. Investigating the role of mitochondrial impairment in AF may guide the path towards new therapeutic and diagnostic targets.	A review of the molecular mechanisms that drive AF and the genesis of this clinical situation.	Novel findings show a key role for mitochondrial dysfunction in the onset and progression of AF. Current therapeutic strategies for AF are aimed directly at the suppression of AF symptoms but are not effective in terms of preventing AF progression. Furthermore, no AF-specific biomarkers are available. So far, several mitochondrial biomarkers have been tested in clinical AF. Recent findings indicate the potential diagnostic value of blood-based 8-OHdG and cfc-mtDNA in staging AF.
Kim, Y. H., 2003 [35]	Korea	Gene expression profiling of oxidative stress on atrial fibrillation in humans	Case-control	Human myocardial tissues under AF and oxidative stress	The study aims to examine the gene transcriptional profiles in human myocardial tissues under AF and oxidative stress conditions.	Researchers studied the effects of oxidative stress on atrial fibrillation (AF) using radioactive DNA microarrays to evaluate changes in gene expression in 26 AF patients undergoing the Maze procedure.	Gene expression profiles reveal 30 upregulated and 25 downregulated genes in AF patients, with five ROS-related genes increasing by over 2.0 and two antioxidant genes decreasing.
Van Wagoner, D. R, 2008 [36]	USA	Oxidative stress and inflammation in atrial fibrillation: role in pathogenesis and potential as a therapeutic target	Review	Atrial fibrillation	The study explores the potential of therapeutic intervention in manipulating oxidative and inflammatory pathways.	This review examines the various treatments for AF utilizing ion channel blockade and the fundamental mechanisms that underlie AF.	AF is a multifactorial arrhythmia characterized by rapid atrial contractile activity degradation, blood stasis, thrombus formation, and increased electrical instability due to aging and other risk factors.
Liu 2015 [37]	China	Serum levels of nicotinamide adenine dinucleotide phosphate oxidase 4 are associated with non-valvular atrial fibrillation.	Clinical study	Atrial fibrillation	The present study aimed to investigate the potential association between serum levels of NOX4 and inflammatory biomarkers and AF.	The study involved 180 consecutive AF patients admitted to Tianjin Medical University’s Department of Cardiology between 2012 and 2013, with a final population of 108 consecutive AF patients.	The study indicates a correlation between elevated NOX4 levels and AF, indicating a potential role of NOX4 in the pathophysiology of human AF.
Cangemi, 2009 [38]	Italy	Different behavior of NOX2 activation in patients with paroxysmal/persistent, or permanent atrial fibrillation	Case-control	Patients with atrial fibrillation and the Patient Control Group.	To define the role of NOX2 and isoprostanes, a marker of oxidative stress, in the different settings of AF.	A study involving 174 patients with AF and 90 controls was conducted, measuring urinary isoprostanes and serum levels of soluble NOX2-derived peptides.	Patients with paroxysmal/persistent AF had higher urinary isoprostane and sNOX2-dp concentrations than permanent AF and controls, with baseline values independently associated with AF type.
David Bode 2021 [39]	Germany	Implications of SGLT Inhibition on Redox Signaling in Atrial Fibrillation	Review	Atrial fibrillation	The study evaluates the implications of SGLT inhibition on redox signaling in AF.	This study involved a review of clinical data and trials, exploring the association between diabetes, AF, and the use of SGLTi.	The study findings indicate that SGLTi has potential benefits in reducing AF burden.
Ryo Nishinarita, 2021 [40]	Japan	Canagliflozin Suppresses Atrial Remodeling in a Canine Atrial Fibrillation Model	Experimental study	Beagle dogs undergoing rapid atrial pacing	The study explored the potential benefits of Canagliflozin (CAN) and other SGLT2 inhibitors in preventing atrial fibrillation (AF) and inhibiting atrial remodeling promotion.	The study involved 12 beagle dogs, 10 undergoing rapid atrial pacing, and compared their performance over three weeks, analyzing parameters and histological findings.	CAN treatment reduces atrial-effective refractory period, conduction velocity, and atrial artery disease (AF) incidence, mitigating interstitial fibrosis and oxidative stress in atrial tissues.
Koizumi; 2023 [41]	Japan	Empagliflozin suppresses mitochondrial reactive oxygen species generation and mitigates the inducibility of atrial fibrillation in diabetic rats.	Experimental study	Type-2 diabetic rat’s atrium	The study investigated the potential of empagliflozin to reduce mitochondrial reactive oxygen species (ROS) generation and reduce fibrosis in diabetic patients, considering the correlation between oxidative stress and AF pathogenesis.	The study examined the effects of empagliflozin on atrial mitochondrial respiratory capacity, reactive oxygen species generation, oxidative stress markers, protein expression, atrial tachyarrhythmia inducibility, and fibrosis in a type-2 diabetes model.	The study suggests empagliflozin may have cardioprotective effects by reducing mitochondrial ROS generation in diabetic rats’ atrium, potentially suppressing the development of atrial fibrillation (AF) in type-2 diabetes.
Aragón- Herrera, 2022 [42]	Spain	Relaxin-2 plasma levels in atrial fibrillation are linked to inflammation and oxidative stress markers.	Case-control	Caucasian patients with persistent AF and Patient Control Group	The study investigates the correlation between relaxin-2 plasma levels in the left atrium and peripheral vein with fibrosis, inflammation, and oxidative stress in AF patients and its anti-fibrotic properties.	The study involved 68 Caucasian patients with persistent AF who underwent pulmonary vein radiofrequency catheter ablation at the University Clinical Hospital of Santiago de Compostela.	Patients with higher relaxin-2 concentrations in peripheral veins had higher Gal-3 levels in plasma, and RLX2 treatment reduced mRNA expression levels in NHCF-A cells.
Liu, 2023 [43]	China	Costunolide ameliorates angiotensin II-induced atrial inflammation and fibrosis by regulating mitochondrial function and oxidative stress in mice: a possible therapeutic approach for atrial fibrillation.	Experimental study	Male C57BL/6 mice induced with AF	The study examines the positive impact of costunolide on angiotensin 2-induced atrial fibrillation.	Male C57BL/6 mice induced with AF using Ang II were administered varying doses of Costunolide (COS) for three weeks.	Costunolide has shown potential therapeutic benefits in treating Angiotensin II-induced atrial fibrillation by reducing inflammation, fibrosis, and mitochondrial dysfunction.
Xu, 2021 [44]	Japan	The xanthine oxidase inhibitor febuxostat reduces atrial fibrillation susceptibility by inhibiting oxidized CaMKII in Dahl salt-sensitive rats.	Experimental study	Dahl salt-sensitive rats	The study evaluated the impact of febuxostat, an XO inhibitor, on salt-induced hypertension in a rat model compared to allopurinol.	Researchers studied Dahl salt-sensitive rats on high-salt diets, dividing them into three groups and administering treatments orally. They measured blood pressure, atrial fibrillation, and protein expression.	Febuxostat and allopurinol significantly reduced hypertension-related atrial fibrillation in rats, improving calcium handling. XO inhibitors reduced Ca^2+^ handling protein expression and partially restored connexin 40 expressions.
Yong-Yan Fan; 2019 [45]	China	Effects of febuxostat on atrial remodeling in a rabbit model of atrial fibrillation induced by rapid atrial pacing	Experimental study	Rabbits with different RAP levels	The study evaluated the impact of febuxostat on atrial remodeling in a rabbit model of atrial fibrillation (AF) induced by rapid atrial pacing and explored its mechanisms.	Rabbits were divided into four groups, each with different RAP levels. The effects of febuxostat on atrial remodeling, inflammation, oxidative stress markers, and left atrium signaling pathways were examined.	Rapid atrial pacing in rabbits leads to atrial enlargement, dysfunction, and fibrillation, while Febuxostat treatment suppresses these changes by inhibiting atrial electrical and structural remodeling.
Gong; 2020 [46]	China	Wenxin Keli Regulates Mitochondrial Oxidative Stress and Homeostasis and Improves Atrial Remodeling in Diabetic Rats	Experimental study	Atrial fibroblasts isolated from neonatal Sprague-Dawley (SD) rats	This study evaluated the hypothesis that WXKL can improve atrial remodeling in diabetic rats, restoring mitochondrial function.	Primary cultures of atrial fibroblasts isolated from neonatal Sprague-Dawley (SD) rats were used. Male SD rats were divided into control, DM (diabetes mellitus), and DM + WXKL (WXKL treatment) groups. Diabetes induction was performed by injection of STZ, followed by treatment with WXKL.	WXKL prevents oxidative stress and improves mitochondrial function. In diabetic rats treated with WXKL, several parameters were improved, including atrial fibrosis, reduced atrial diameter, increased atrial conduction velocity, and reduced induction of atrial fibrillation.
YU, 2023 [47]	China	Andrographolide protects against atrial fibrillation by alleviating oxidative stress injuries and promoting impaired mitochondrial bioenergetics	Experimental study	HL-1 cells and rabbits	This study aimed to explore the mechanisms of action of andrographolide on AF.	The study investigated Andr’s role in atrial fibrillation (AF) by pre-treating HL-1 cells and rabbits with Andr before RES and atrial pacing using RNA sequencing.	Andrographolide effectively mitigates rapid atrial pacing, causing changes in electrophysiology, inflammation, oxidative damage, and apoptosis, potentially through a therapeutic mechanism involving the Keap1-Nrf2 complex.
Bukowska, 2007 [48]	Germany	Mitochondrial dysfunction and redox signaling in atrial tachyarrhythmia	Case-control	Ex vivo atrial tissue from patients with and without atrial fibrillation	The study investigates the impact of AF on mitochondrial dysfunction and oxidative stress-activated signal transduction by analyzing NF-kB, LOX-1, ICAM-1, and HO-1.	Ex vivo atrial tissue from patients with and without atrial fibrillation was studied for mitochondrial structure and respiration, while NF-kB target gene expression was measured using various methods.	Oxidative stress, mitochondrial structure, and respiration were observed in human atrial tissue slices, with NF-jB accumulation and elevated ICAM-1 expression. A blockade with verapamil prevented these changes.
Hadi, 2010 [49]	United Arab Emirates	Inflammatory cytokines and atrial fibrillation: current and prospective views	Post-hoc comparison of data collected in a prospective randomized investigation	Atrial fibrillation	To present an overview of the evidence linking inflammatory cytokines to AF.	The authors analyzed articles published until December 2009 on Medline, Pubmed, Scopus, and EBSCOhost^®^ using indexing terms for inflammation, cytokines, AF, and atrial arrhythmias.	Inflammatory cytokines and markers like IL-6 and CRP are linked to atrial fibrillation (AF), potentially indicating inflammation and predicting thromboembolic events in AF patients.
Zhao, 2020 [50]	China	Attenuation of atrial remodeling by aliskiren via affecting oxidative stress, inflammation, and the PI3K/Akt signaling pathway	Experimental study	Dogs subjected to rapid atrial pacing	The study investigates the cardioprotective effect of aliskiren (ALS) and its potential molecular mechanisms in atrial remodeling, focusing specifically on atrial fibrillation (AF).	The study involved acute and chronic experiments on dogs subjected to rapid atrial pacing, assessing parameters like effective refractory periods, AF inducibility, and average duration.	Aliskiren has shown cardioprotective effects by reducing electrophysiological alterations, oxidative stress, inflammation, and atrial remodeling, possibly through regulating the PI3K/Akt signaling pathway.
Xue, 2020 [51]	China	Exogenous hydrogen sulfide reduces atrial remodeling and atrial fibrillation induced by diabetes mellitus via activation of the PI3K/Akt/eNOS pathway	Experimental study	Sprague-Dawley rats	This study aimed to explore the impact of hydrogen sulfide on diabetes mellitus-induced atrial fibrillation and its underlying mechanisms.	The study involved Sprague-Dawley rats in four groups: control, DM, H2S, and DM + H2S, analyzing atrial fibrillation, fibrosis, protein expression, cell viability, and cardiac fibroblast migration.	H2S may reduce atrial fibrosis and DM-induced AF by activating the PI3K/Akt/eNOS pathway.
Li, 2010 [52]	EUA	Role of inflammation and oxidative stress in atrial fibrillation	Case-control	Patients with and without AF	The study aims to understand the role of inflammation and oxidative stress in developing AF.	A study compared 305 AF patients with and without AF, assessing serum inflammatory markers and oxidative stress and comparing them to control patients.	IL-6, IL-8, IL-10, TNF-α, MCP1, VEGF, and NTpBNP concentrations were linked to AF, with graded increases in TNF-α and NTpBNP among paroxysmal, persistent, and permanent AF subgroups.
Andelova, 2022 [53]	Slovakia	Mechanisms Underlying the Antiarrhythmic Properties of Cardioprotective Agents Impacting Inflammation and Oxidative Stress	Review	Atrial fibrillation	The study aimed to examine the antiarrhythmic efficacy and molecular mechanisms of current clinically used pharmaceuticals in the context of AF.	The study examines the biomolecular mechanisms of FA and the therapeutic efficacy of Sodium-Glucose Cotransporter-2 Inhibitors, Statins, and Omega-3 fatty acids in preventing oxidative and inflammatory stress.	The approach suggests that reducing oxidative and inflammatory stress can eliminate pro-arrhythmic factors and arrhythmia substrates, making it a potent tool for reducing cardiac arrhythmia burden.
Han, 2008 [54]	China	Nitric oxide overproduction derived from inducible nitric oxide synthase increases cardiomyocyte apoptosis in human atrial fibrillation	Case-control	patients with permanent AF and sinus rhythm after mitral valve replacement surgery	The study aims to investigate the potential role of iNOS in atrial remodeling in AF.	The study investigated patients with permanent AF and sinus rhythm after mitral valve replacement surgery, using Western blotting, immunohistochemical staining, and the NOX Detection Kit to measure cardiac function.	NOS expression in the right atrium was upregulated, while eNOS did not. Inflammation and oxidative damage in the right atrium of AF patients were associated with increased iNOS/eNOS expression.
Bukowska, 2010 [55]	Germany	Atrial expression of endothelial nitric oxide synthase in patients with and without atrial fibrillation	Case-control	atrial tissue from 234 patients with atrial fibrillation	The study aims to assess the endocardial expression of eNOS in atrial tissue samples from patients with and without atrial fibrillation (AF).	Tissue microarrays were used to analyze atrial tissue from 234 patients, examining differences in atrial eNOS expression, with immunohistological results confirmed by Western blotting in selected patients.	eNOS expression is influenced by factors like diabetes mellitus and coronary artery disease, with women with AF having the lowest levels.
Neuman, 2007 [56]	USA	Oxidative stress markers are associated with persistent atrial fibrillation	Cross-sectional study	males with or without AF	To compare serum markers of oxidation and associated inflammation in individuals with or without AF.	A cross-sectional study compared serum markers of oxidative stress and inflammation in 40 males with or without AF, matched by age, sex, diabetes, and smoking status.	Oxidative stress, not inflammatory markers, is statistically associated with AF, suggesting that oxidative stress markers may be predictive in AF management.
Pinho- Gomes, 2014 [57]	United Kingdom	Targeting inflammation and oxidative stress in atrial fibrillation: Role of 3-hydroxy-3-methylglutaryl-coenzyme reductase inhibition with statins	Review	Atrial fibrillation	Using statins to decrease inflammation by restoring the myocardial nitroso-redox balance.	This review explores articles discussing potential statin-related therapies and their potential to reduce inflammation.	Statins show the highest anti-arrhythmic benefits in preventing postoperative AF but limited benefits in primary AF prevention, making them unsuitable for preventing incident AF or recurrence.
Godoy- Marín, H.; 2021 [58]	Spain	Adenosine a2a receptors are upregulated in peripheral blood mononuclear cells from atrial fibrillation patients	Case-control	Samples from patients with sinus rhythms and atrial fibrillation	The study explores the expression of adenosine A2A receptor (A2AR) in right atrium biopsies and peripheral blood mononuclear cells from non-dilated sinus rhythm (ndSR), dilated sinus rhythm (dSR), and AF patients.	Samples from patients with sinus rhythms and atrial fibrillation were collected and analyzed using various methods, including gel electrophoresis, immunoblotting, RT-qPCR, cell culture, Flow Cytometry, and Confocal Imaging.	The study found increased A2AR expression in the right atrium of AF patients, with adenosine content and reduced ADA activity in plasma, and a positive correlation between A2AR expression and PBMCs.
Avula, 2021 [59]	USA	Attenuating persistent sodium current-induced atrial myopathy and fibrillation by preventing mitochondrial oxidative stress	Experimental study	crossbreeding mice expressing persistent sodium channels and mice expressing human mitochondrial catalase (mCAT)	This study aims to comprehend the mechanisms influencing structural and electrophysiological remodeling in the atria due to an increased persistent sodium current.	The study involved crossbreeding mice expressing persistent sodium channels (NaV1.5 F1759A) with mice expressing human mitochondrial catalase (mCAT).	mCAT expression reduced mitochondrial oxidative stress, atria structural changes, atrial fibrillation, and ryanodine receptor dysfunction, reducing spontaneous and stimulation-induced atrial fibrillation.
Lin, 2003 [60]	China	Oxidative damage to mitochondrial DNA in the atrial muscle of patients with atrial fibrillation	Case-control	patients with chronic AF	The authors utilized long-range polymerase chain reaction (PCR) to detect large-scale deletions of mtDNA in the atrial muscle of AF patients.	Right atrial appendages were removed from patients with chronic AF and sinus rhythm during heart surgery, and cellular DNA was extracted, revealing large-scale deletions.	The study found a high frequency of large-scale mtDNA deletions, particularly the 4977-bp deletion, in patients with atrial fibrillation (AF), with oxidative damage causing more significant damage.
Istratoaie, 2022 [61]	USA	Paraoxonase 1 and atrial fibrillation: Is there a relationship?	Case-control	patients with symptomatic paroxysmal or persistent AF and the patient control group	The study aims to assess the concentration and activity of PON1 arylesterase (AREase) in patients with AF.	The study analyzed 67 patients with symptomatic paroxysmal or persistent AF admitted for cardioversion and 59 without AF, evaluating clinical parameters, lipid profile, PON1 concentration, and AREase.	Oxidative stress contributes to diseases like arrhythmias and increased risk of atrial fibrillation (AF), promoting endocardial dysfunction, left atrial thrombus, and stroke and influencing the efficacy of various drugs.
Samman, 2017 [62]	USA	Association between oxidative stress and atrial fibrillation	Prospective study	coronary angiography patients	The study hypothesized that prevalent and incident AF are associated with glutathione (EhGSH) and cysteine redox potentials, estimating systemic oxidative stress.	Aminothiol plasma levels in 1439 coronary angiography patients, including 148 with AF diagnoses, showed an 11.5% incidence of AF in 104 out of 917 patients after 6.3 years.	EhGSH levels in CAD patients increase the risk of prevalent and incident AF, independent of hsCRP level and other AF predictors, and correlate with the CHA2DS2-VASc score.

AF: Atrial Fibrillation; EhGSH: glutathione; CAD: Coronary Arterial Disease; hsCRP: high-sensitivity C-reactive protein; PCR: polymerase chain reaction; PON1: Paraoxonase 1; OXPHOS: Mitochondrial Oxidative Phosphorylation; eNOS: endothelial Nitric Oxide Synthase; ETC: Electron Transport Chain; O_2_: oxygen; NADPH: Nicotinamide Adenine Dinucleotide Phosphate; mtDNA: mitochondrial DNA; ROS: Reactive Oxygen Species; mCAT: mitochondrial catalase; ATP: Adenosine Triphosphate; RAS: Renin-Angiotensin System; EAT: Epicardial Adipose Tissue; 8-OHdG: 8-Hydroxy-2′-deoxyguanosine; NOX: NADPH oxidase; SGLT: sodium-Glucose Linked Transporter; CAN: Canagliflozin; Gal-3: galectin-3; COS: Costunolide; XO: xanthine oxidase; WXKL: Wenxin Keli; DM: diabetes mellitus; Keap1: Kelch-like ECH-Associating Protein 1; Nrf2: Nuclear factor erythroid 2-related factor 2; NF-kB: Nuclear Factor-kappa B; LOX-1: Lectin-like Oxidized Low-Density Lipoprotein Receptor-1; ICAM-1: Adhesion Molecule-1; HO-1: Heme Oxygenase-1; ALS: aliskiren; TNF-α: Tumor Necrosis Factor-alpha; MCP1: Monocyte Chemoattractant Protein-1; VEGF: Vascular Endothelial Growth Factor; NTpBNP: N-terminal pro-B-type; iNOS: inducible Nitric Oxide Synthase; A2AR: adenosine A2A receptor; ndSR: non-dilated sinus rhythm; dSR: dilated sinus rhythm.

**Table 2 ijms-25-00535-t002:** New target molecules for AF treatment.

KL1333	Increases mitochondrial activity and reduces oxidative stress in fibroblasts in patients with mitochondrial encephalomyopathy, lactic acidosis, and stroke-like events. It also increases NAD^+^ levels and stimulates sirtuin 1/AMP-activated protein kinase/peroxisome proliferator-activated receptor-gamma coactivator 1alphasignaling [69].
KH176	By interacting with the thioredoxin system and the enzymatic mechanism of peroxiredoxin, the drug KH176 can effectively reduce elevated cellular levels of reactive oxygen species and protect primary cells deficient in oxidative phosphorylation from redox disorders [27,70].
Ru360	The study by Pool et al. demonstrated that Ru360 prevents mitochondrial overload of Ca^2+^, dysfunction of this organelle, and, consequently, contractile dysfunction. However, it is used only in preclinical settings [34].
Antioxidant SS31	The antioxidant SS31, currently tested in clinical trials, improves the coupling of electron transport chain complexes, and thus enhances mitochondrial bioenergetics and suppresses the abundance of ROS and oxidative stress [34].
NAD^+^ supplementation	It is a possibility for preserving mitochondrial function since homeostasis of NAD^+^ improves function by reducing oxidative stress and DNA damage [34].
L-glutamine	It has nutraceutical potential for the treatment of AF, as it stabilizes the microtubular network, increases the expression of heat shock protein in degenerative and inflammatory diseases, and contributes to the suppression of ROS and DNA damage induced by ROS due to its antioxidant activity [34].

**Table 3 ijms-25-00535-t003:** Therapeutic possibilities and their respective main effects for AF treatment.

Therapeutic Possibilities	Main Effects
Statins	Reduction of C-reactive protein (CRP); prevention of inflammation, consequently preventing electrical and structural remodeling; prevention of oxygen free radical (ROS) synthesis induced by NADPH oxidase.
Steroids	Anti-inflammatory activity, indirect antioxidant, and immunomodulatory properties. Promotes reduction of atrial endothelial protein nitric oxide synthase levels and CRP levels.
Carvedilol	α1 blocking and antioxidant properties and anti-oxidation effects, in addition to exerting modulating effects on ionic channels and currents.
Dipeptidyl Peptidase-4 inhibitors	Reduction of ROS, promotion of mitochondrial oxidative stress, improvement of mitochondrial function, preservation of mitochondrial biogenesis, and reduction of inflammation.
Selective Sodium-Glucose Cotransporter 2 Inhibitors	Reduction of arterial resistance, improving endothelial function; normalization of sodium and calcium cytosolic concentrations; reduction of ROS synthesis; promotion of prevention of atrial remodeling and reduction of atrial fibrillation (AF) burden; promotion of less systemic inflammation; inhibition of atrial fibrosis and cardiomyocyte hypertrophy. In addition, it promotes a 19% reduction in AF in patients with diabetes, regardless of pre-existing AF or heart failure. In addition, they are suspected of promoting the reduction of pro-inflammatory molecules, increasing adiponectin, and suppressing inflammatory markers in the myocardium.
Ubiquinone	Anti-inflammatory antioxidant activity has a beneficial effect on mitochondrial function and significantly suppresses DNA damage.
Thiazolidinediones	Reduction of atrial remodeling. They prevent the recurrence of AF after electrical cardioversion, reduce cardiac risk factors and surrogate indicators of cardiovascular disease, and reduce the frequency of cardiac events in individuals with diabetes.
Trimetazidine	Reduction of ROS synthesis by acting directly on the activity of the respiratory chain. In addition, it prevents structural atrial remodeling, reduces the inducibility of AF, and shortens the duration of AF.
Ranolazine	Reduction of oxidative stress, improvement of mitochondrial function, suppression of apoptosis, and reduction of the likelihood of developing AF by approximately 50%. In addition, it increases the success rate of amiodarone cardioversion.
A diet rich in antioxidants	Vitamins E and C are antioxidants and eliminate ROS, such as O_2_, OH, peroxynitrite, sulfhydryl radicals, and oxidized low-density lipoprotein.
Mitochondrial transcription factor A (TFAM)	It increases ATP content by upregulating NADH-1 mitochondrial-coded dehydrogenase and cytochrome c oxidase-1 mitochondrially-coded expression levels.
Relaxin-2	Reduction of oxidative stress (decrease in plasma levels of hydrogen peroxide and ROS), inhibition of profibrotic molecules, and suppression of inflammation, with a decrease in gene expression of inflammatory markers. In vitro, treatment with relaxin-2 inhibited the migration of normal human atrial cardiac fibroblasts. Furthermore, it reduced mRNA and protein levels of the profibrotic molecule, transforming growth factor-beta1 (TGF-β1).
Costunolide	Reduces inflammation and fibrosis induced by angiotensin II, improves mitochondrial function, alleviates oxidative stress by countering excessive ROS production, and activates the factor-2-related erythroid nuclear signaling pathway.
Febuxostat	Reduces the production of ROS, inhibits xanthine oxidase, and combats oxidative stress and inflammation, showing a decrease in inflammatory markers and the activity of antioxidant enzymes. Additionally, it positively influences AF by regulating the TGF-β1/Smad signaling pathway, which plays a role in collagen production and fibrosis.
Aliskiren	Attenuates electrical and structural atrial remodeling induced by rapid atrial pacing, reducing inflammation and oxidative stress. Furthermore, it regulates the PI3K/Akt signaling pathway.
Wenxin Keli	Antiarrhythmic properties and selective inhibition of atrial sodium current. It improves mitochondrial function by increasing respiration and reducing ROS production. In diabetic rats, Wenxin Keli prevents AF by enhancing atrial remodeling and restoring mitochondrial function.
Hydrogen sulfide	Activation of the PI3K/Akt/eNOS signaling pathway is associated with a reduction in the production of ROS. H_2_S can reduce diabetes-induced AF, decreasing the incidence and persistence of AF without affecting glucose metabolism.
Andrographolide	Reduction of cardiac cell apoptosis, improvement of mitochondrial function, antioxidant role, regulation of calcium homeostasis genes, and influence on transcription factors like factor-2-related erythroid nuclear.
Metformin	Activation of AMPK Src kinase, normalization of connective tissue expression, and prevention of atrial remodeling via the AMPK/PGC-1/PPAR pathway. Preserves mitochondrial function, improving the oxygenation and activity of complexes I, II, and IV. Increases PGC-1 and Coenzyme Q10 expression, providing antioxidant benefits and membrane stabilization.
Fibrates	Impact mitochondrial function through the PPAR/PGC-1 pathway, potentially mitigating metabolic remodeling by regulating the PPAR/sirtuin route 1/PGC-1, thereby reversing the shortening of the atrial refractory period.
Elamipretide	It improves mitochondrial efficiency and reduces the production of ROS by stabilizing the mitochondrial membrane and cytochrome C, increasing ATP production, normalizing the ATP/ADP ratio, and reducing TNF and CRP levels.
Genetic therapy	Restores average heart rate and improves heart rate control in animal models of AF. However, they have not yet reached the phase of widespread clinical use.

## Data Availability

The data supporting this study’s findings are available on request from the corresponding author, (A.d.S.M.J.).

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
