# Peer review of "Developing Pharmacological Therapies for Atrial Fibrillation Targeting Mitochondrial Dysfunction and Oxidative Stress: A Scoping Review"

_ijms, 2023, doi:10.3390/ijms25010535_

Round 1

Reviewer 1 Report

Comments and Suggestions for Authors

I have reviewed the manuscript entitled "Developing Pharmacological Therapies for Atrial Fibrillation Targeting Mitochondrial Dysfunction and Oxidative Stress: A scoping Review" By Antonio da Silva Menezes Junior et al. In this review, the authors really dig into all pathways related to ROS activation and how that affects mitochondrial function and leads to disturbed electrical pathways in the atria as well as contribute to the remodeling process and permanent AF. This is an extremely solid review article, which I truly enjoyed reading. As I can see, this is the first of its kind that take this broad picture of oxidative stress/ ROS and AF induction. There are some issues that I would like you to address:

1) The text in the figures are not clear (blur), please fix this.

2) Table 2, Relaxin-2 is under-lined- any reason for this?

3) In the introduction, page 3, one paragraph starts with: "Antiarrhythmic agents include pharmacological or radiofrequency catheter ablation, which uses rate- or rhythm-control agents". I think it should be rephrased:

Antiarrhythmic therapies include pharmacological agents for rhytm and rate control and cather-based interventions, which mainly are effective on paroxysmal AF and less on permanent AF. 

Author Response

Dear Reviewer 1,

I would like to express my sincere appreciation for your valuable insights and suggestions during the review of my paper titled "Developing Pharmacological Therapies for Atrial Fibrillation Targeting Mitochondrial Dysfunction and Oxidative Stress: A Scoping Review." Your thorough analysis and feedback are crucial for enhancing the quality of my work.

We have carefully considered all your comments and suggestions. Below, I provide our responses to each of the points raised:

  1. We acknowledge your concern regarding the clarity of text in the figures, and we apologize for any inconvenience caused. We assure you that we are committed to addressing this issue promptly and enhancing the overall quality of the figures.

Configuration Change: The file format of the images has been updated from JPEG to TIFF.

Resolution Adjustment: The resolution of the images has been increased from 300 dpi to 600 dpi.

  1. Regarding your observation about Table 2, specifically the underlining of "Relaxin-2," we want to clarify that the underlining was unintentional, and we have taken immediate steps to rectify this issue. In the revised version of the manuscript, the underlining has been removed to ensure consistency and clarity.
  2. Regarding your comment on the paragraph in the introduction (page 3) that begins with "Antiarrhythmic agents include pharmacological or radiofrequency catheter ablation, which uses rate- or rhythm-control agents," we have carefully reviewed your suggestion and made the necessary alterations. The revised paragraph now reads as follows:
  •  
  • "Antiarrhythmic therapies encompass pharmacological agents designed for both rhythm and rate control, along with catheter-based interventions. The latter primarily demonstrate efficacy in paroxysmal atrial fibrillation (AF) and exhibit comparatively less impact on permanent AF."

Additionally, I would like to emphasize our commitment to making a meaningful contribution to the advancement of knowledge in the field addressed by our paper. We are willing to make further adjustments as necessary to ensure that the content is clear, precise, and impactful.

Sincerely,

Antonio da Silva Menezes Junior MD, Ph.D. 

Dear Reviewer 1,

I would like to express my sincere appreciation for your valuable insights and suggestions during the review of my paper titled "Developing Pharmacological Therapies for Atrial Fibrillation Targeting Mitochondrial Dysfunction and Oxidative Stress: A Scoping Review." Your thorough analysis and feedback are crucial for enhancing the quality of my work.

We have carefully considered all your comments and suggestions. Below, I provide our responses to each of the points raised:

  1. We acknowledge your concern regarding the clarity of text in the figures, and we apologize for any inconvenience caused. We assure you that we are committed to addressing this issue promptly and enhancing the overall quality of the figures.

Configuration Change: The file format of the images has been updated from JPEG to TIFF.

Resolution Adjustment: The resolution of the images has been increased from 300 dpi to 600 dpi.

  1. Regarding your observation about Table 2, specifically the underlining of "Relaxin-2," we want to clarify that the underlining was unintentional, and we have taken immediate steps to rectify this issue. In the revised version of the manuscript, the underlining has been removed to ensure consistency and clarity.
  2. Regarding your comment on the paragraph in the introduction (page 3) that begins with "Antiarrhythmic agents include pharmacological or radiofrequency catheter ablation, which uses rate- or rhythm-control agents," we have carefully reviewed your suggestion and made the necessary alterations. The revised paragraph now reads as follows:
  •  
  • "Antiarrhythmic therapies encompass pharmacological agents designed for both rhythm and rate control, along with catheter-based interventions. The latter primarily demonstrate efficacy in paroxysmal atrial fibrillation (AF) and exhibit comparatively less impact on permanent AF."

Additionally, I would like to emphasize our commitment to making a meaningful contribution to the advancement of knowledge in the field addressed by our paper. We are willing to make further adjustments as necessary to ensure that the content is clear, precise, and impactful.

Sincerely,

Antonio da Silva Menezes Junior MD, Ph.D. 

Reviewer 2 Report

Comments and Suggestions for Authors

The authors aimed to elucidate the role of mitochondrial oxidative mechanisms in AF pathophysiology, the impact of mitochondrial oxidative stress on
AF initiation and perpetuation, and current therapies. They concluded that oxidative stress and inflammation are intrinsically linked, and inflammatory pathways are highly correlated with the occurrence of AF.

I have the following concerns:

1. What kind of studies were included in the review?

2.  What were the exclusion criteria?

3. What kind of atrial fibrillation patterns were included?

4. I suggest to assess the risk of bias

5. Please provide the baseline characteristics of the studied population

Comments on the Quality of English Language

Minor English edits required

Author Response

Rebuttal Letter 2

23.12.2023

Dear Reviewer 2,

I would like to express my sincere appreciation for your valuable insights and suggestions during the review of my paper titled "Developing Pharmacological Therapies for Atrial Fibrillation Targeting Mitochondrial Dysfunction and Oxidative Stress: A Scoping Review." Your thorough analysis and feedback are crucial for enhancing the quality of my work.

We have carefully considered all your comments and suggestions. Below, I provide our responses to each of the points raised:

  1. The review encompassed a variety of study designs, including case-control studies, reviews, clinical studies, experimental studies, randomized controlled trials, cross-sectional studies, and prospective studies.
  2. The exclusion criteria encompassed secondary sources, including editorials, books, expert opinion articles, dissertations, theses, and conference abstracts, except literature reviews, which were deliberately included in the review.
  3. In our study, the specific patterns of atrial fibrillation were not delineated or discriminated.
  4. Thank you for your suggestion to assess the risk of bias. According to the PRISMA ScR guidelines, the concept of "risk of bias" is more commonly applied to systematic reviews of interventions. In the context of a scoping review, which encompasses diverse sources of evidence such as quantitative and/or qualitative research, expert opinion, and policy documents, the recommended approach is to employ the methods of Critical Appraisal of Individual Sources of Evidence and Summary of Evidence. These methodologies are in alignment with the PRISMA ScR guidelines, as outlined in the work by Tricco et al., titled "PRISMA Extension for Scoping Reviews (PRISMAScR): Checklist and Explanation" (Ann Intern Med. 2018;169:467–473, doi: 10.7326/M18-0850).
  5. The baseline characteristics of the studied population, originally presented in Supplementary Table 1, have been repositioned within the main body of the text for enhanced reader comprehension. This information is now incorporated as Table 1 in the revised manuscript for improved accessibility and clarity.
  6. We have used Editage to modify the English text to clarify the grammar and spelling as if we were native speakers.

Additionally, I would like to emphasize our commitment to making a meaningful contribution to the advancement of knowledge in the field addressed by our paper. We are willing to make further adjustments as necessary to ensure that the content is clear, precise, and impactful.

Round 2

Reviewer 2 Report

Comments and Suggestions for Authors

Thank you for your revision. I have no further comments

Comments on the Quality of English Language

Minor English edits required.